# ENERGY GUIDED SMOOTHNESS TO IMPROVE ROBUSTNESS

## ABSTRACT

Graph Neural Networks (GNNs) perform well on graph classification tasks but are notably susceptible to label noise, leading to compromised generalization and overfitting. We investigate GNNs' robustness, identify generalization failure modes and causes, and prove our hypothesis with three robust GNN training methods. Specifically, GNN generalization is compromised by label noise in simpler tasks (few classes), low-order graphs (few nodes), or highly parameterized models. Focusing on graph classification, we show the link between GNN robustness and the smoothness of learned node representations, as quantified by the Dirichlet energy. We show that GNN learns smoother representations with decreasing Dirichlet energy during training, until the model fits on noisy labels, adding high-frequency components to the representations. To verify our analysis, we propose three robust training strategies for GNNs: (a) a spectral inductive bias by enforcing positive eigenvalues in GNN weight matrices to demonstrate the link between smoothness and robustness; (b) a Dirichlet energy overfitting control mechanism, which relies on a noise-free validation set; (c) a noise-robust loss function tailored for GNNs to induce smooth representations. Crucially, our methods do not degrade performance in noise-free data, reinforcing our central hypothesis that GNNs' smoothness bias defines their robustness to label noise.

## 1 INTRODUCTION

Graph Neural Networks (GNNs) are powerful for modeling graph structured data Zhang et al. (2018), especially for solving the graph classification task where the objective is to assign a label to each graph in the dataset. The applications of graph classification span across domains, including social and citation networks Yanardag & Vishwanathan (2015a), bioinformatics Borgwardt et al. (2005), and chemical molecule analysis (Wale & Karypis, 2006). Crucially, in many real world applications, the label acquisition process is noisy. Compared to image Algan & Ulusoy (2021) or node classification Dai et al. (2021), the problem of graph classification with noisy labels is relatively less explored. While initial pioneering works Nt & Maehara (2019); Yin et al. (2023) have begun to address this challenge, a systematic understanding of when and why GNNs are vulnerable to noisy labels and the development of robust mitigation strategies tailored to the unique inductive biases of GNNs remain active areas of research. In this paper, we address this noise robustness challenge and study GNNs' susceptibility to label noise when some samples are labeled incorrectly. The conventional understanding is that cross entropy loss (CE), usually used in GNN classification tasks, typically leads to overfitting in the presence of noisy labels, particularly when the model has sufficient expressivity (Zhang et al., 2017a). We observed that, for graph classification, GNNs trained with standard CE can show varying degrees of robustness when exposed to label noise. We investigate the hypothesis that GNNs leverage smooth representations as an inductive bias for generalization and noise robustness. Through theoretical and empirical analysis, we confirm this link and show that noise overfitting corresponds to latent node features sharpening. Based on these insights, we develop methods to detect overfitting and propose three distinct strategies to enhance robustness in noisy graph classification. Specifically, our contributions are the following:

- We present the first systematic study linking **label noise robustness in graph classification** to the **spectral dynamics of Dirichlet energy ($E^{\text{dir}}$)**. While prior works have studied oversmoothing and energy decay in node classification, we reveal how noise memorization

in graph classification corresponds to a characteristic rise in high frequency Dirichlet components.

- We propose a **unifying energy based perspective** on robustness, showing that three seemingly different approaches (i) enforcing positive eigenvalues in GNN weights, (ii) directly regularizing Dirichlet energy, and (iii) introducing the novel GCOD loss can all be understood as mechanisms that constrain harmful high frequency energy.

- We provide **comprehensive empirical evidence** across diverse benchmarks and both symmetric and asymmetric noise, establishing Dirichlet energy as a reliable signal of overfitting. Crucially, our methods **improve robustness without degrading clean data performance**.

Together, these contributions introduce a principled framework that connects spectral smoothness, Dirichlet energy, and noise robust learning in GNNs. We believe this perspective opens a new direction for designing graph models that are not only robust to label noise, but also more stable under domain shift and adversarial perturbations. Overall, this study improves our understanding of the sources of GNNs robustness, its smoothness, and its inductive bias, and offers guidance for practitioners to apply GNNs efficiently to real world applications. The code is available at `https://anonymous.4open.science/r/Robustness_Graph_Classification-E76F`.

## 2    RELATED WORKS

**Learning under label noise.** A large body of work is devoted to the challenge of learning with noisy labels. Several methods are based on **robust loss functions**, using symmetric losses Ghosh et al. (2017), or loss correction methods (Patrini et al., 2017). Other works are the **Neighboring-based noise identification** approaches (Zhu et al., 2022). Several approaches are based on the **early learning phenomenon** Arpit et al. (2017), and others proposed to improve the quality of training data by treating samples with a small loss value as correctly labeled during the training process (Gui et al., 2021). Additional methods for learning in the presence of noisy labels are in Appendix D.

**Graph Learning in noisy scenarios.** The methods discussed earlier focus mainly on learning from noisy labels in image datasets. Unlike images, graphs exhibit noise in labels, graph topology (e.g., adding/removing edges or nodes), and node features. Most prior work discusses noisy node labels NT et al. (2019); Yuan et al. (2023a); Yin et al. (2023); Yuan et al. (2023b); Li et al. (2024); Dai et al. (2021); Kang et al. (2018), while noise at the edge and feature levels have also been explored (Fox & Rajamanickam, 2019; Dai et al., 2022; Yuan et al., 2023b). However, fewer studies investigate graph classification under noise, limiting progress in applying and improving graph classification tasks. The seminal work of NT et al. (2019) addresses graph classification with label noise, proposing a surrogate loss to discard noisy labels under certain assumptions, without comparison to clean label scenarios. More recently, Yin et al. (2023) introduces a method combining contrastive learning and MixUp Lim et al. (2021) within the loss function to improve generalization, along with a curriculum learning strategy to dynamically discard noisy samples. In contrast, we propose tackling noisy labels using an effective loss function inspired by Wani et al. (2024) or by enhancing graph smoothness.

**Dirichlet Energy, Smoothing bias and Sharpening** Dirichlet energy is a key measure in GNN, quantifying the smoothness or variation of features across nodes (Zhou & Schölkopf, 2005). Most GNNs function as low pass filters, emphasizing low frequency components while diminishing high frequency ones (Nt & Maehara, 2019; Rusch et al., 2023). Specifically, Nt & Maehara (2019) showed this phenomenon holds for graphs without non trivial bipartite components, with self loops further shrinking eigenvalues. Cai & Wang (2020); Oono & Suzuki (2021) prove that GNN Dirichlet energy exponentially decreases with additional layers when the product of the largest singular value of the weight matrix and the largest eigenvalue of the normalized Laplacian is less than one. GNN learnable weight matrices fundamentally control whether features are smoothed or sharpened (Di Giovanni et al., 2023). While Dirichlet energy evolution has been studied in relation to oversmoothing Cai & Wang (2020); Nt & Maehara (2019), and various mitigation approaches leveraging energy properties exist Bo et al. (2021); Zhou et al. (2021a); Chen et al. (2023), these work has primarily focused on node classification, where oversmoothing significantly impacts performance (Yan et al., 2022). The role of energy dynamics in graph classification, especially with label noise, remains less explored. In this work, we provide theoretical and practical insights on leveraging Dirichlet energy to enhance graph classification performance, even in the presence of label noise (comprehensive discussion in Appendix D).

# 3 BACKGROUND

Let $\mathcal{G} = (\mathcal{V}, \mathcal{E}, \mathbf{X})$ be an undirected graph, with $\mathcal{V}$ the set of nodes and $\mathcal{E}$ the set of edges. We denote by $N = |\mathcal{V}|$ the number of nodes of $\mathcal{G}$. $\mathcal{N}_u$ is the neighborhood of the node $u$, and $d_u = |\mathcal{N}_u|$ is its degree. $\mathbf{D} \in \mathbb{R}^{N \times N}$ is the degree matrix, a diagonal with entries $D_{uu} = d_u$. Each node $u$ has feature vector $\mathbf{x}_u \in \mathbb{R}^m$. The feature matrix $\mathbf{X} \in \mathbb{R}^{N \times m}$ stacks all the feature vectors. $\mathbf{A} \in \{0,1\}^{N \times N}$ is the graph's adjacency matrix, with $A_{uv} = 1$ if $(u,v) \in \mathcal{E}$ and $A_{uv} = 0$ otherwise.

**Graph Neural Networks for Graph Classification.** In graph classification, each sample in the dataset, $\mathcal{D}$, is a graph, i.e., $\mathcal{D} = \{\mathcal{G}^i, \mathbf{y}_i\}_{i=1}^n$, where $\mathcal{G}^i = (\mathcal{V}^i, \mathcal{E}^i, \mathbf{X}^i)$, and $\mathbf{y}_i \in \{0,1\}^{|C|}$ is its class associated one-hot encoded representation. We represent the set of labels for $\mathcal{D}$ as $\mathbf{y} \in \{0,1\}^{n \times |C|}$. More simply we use $c_i$ to express the class of sample $i$. $\mathbf{X}^i \in \mathbb{R}^{N_i \times m}$, and $\mathbf{A}^i \in \mathbb{R}^{N_i \times N_i}$ are the feature and adjacency matrices of graph $i$ respectively. In the case of learning under label noise, in the training data $c_i$ may differ from the ground truth. In this setting, GNNs are employed to extract features from graph structured data. In the message passing formalism Gilmer et al. (2017), each feature matrix $\mathbf{X}^i \ \forall i \in \{1, ..., n\}$ is iteratively updated within the GNN, yielding a new set of latent features $\mathbf{Z}^i \in \mathbb{R}^{N_i \times m'}$ for the graph $\mathcal{G}^i$. We denote the intermediate representations as $\mathbf{H}_i^l$, $0 \leq l \leq L$, for each GNN layer up to the $L$-th one. We identify $\mathbf{X}^i \equiv \mathbf{H}_i^0$ and $\mathbf{Z}^i \equiv \mathbf{H}_i^L$. Given a set of weights $\mathbf{W}_l$ and $\mathbf{\Omega}_l$ for layer $l$, the message-passing update rule for graph $i$ is:

$$\mathbf{H}_i^{l+1} = UP_{\mathbf{\Omega}_l}\left(\mathbf{H}_i^l, \ AGGR_{\mathbf{W}_l}\left(\mathbf{H}_i^l, \ \mathbf{A}^i\right)\right), \quad 0 \leq l \leq L, \ l \in \mathbb{N}, \tag{1}$$

where $UP_{\mathbf{\Omega}_l}$ and $AGGR_{\mathbf{W}_l}$ denote the *update* and *aggregation* functions of the message passing mechanism. After obtaining the final node representations $\mathbf{Z}^i \in \mathbb{R}^{N_i \times m'}$, a learnable, permutation-invariant function $f_\theta : \mathbb{R}^{N_i \times m'} \to \mathbb{R}^{|C|}$ is applied to transform them into class probabilities. The predicted output is then represented as a one hot encoded vector $\hat{\mathbf{y}}_i$.

**Dirichlet Energy on graphs.** We now define the Dirichlet energy for graph data: $E^{dir}$, which quantifies the smoothness of a scalar or vector field defined over the nodes of a graph. For a graph $\mathcal{G}^i = (\mathcal{V}^i, \mathcal{E}^i)$ with node representation matrix $\mathbf{Z}^i \in \mathbb{R}^{N_i \times m'}$, where $\mathbf{Z}^i$ denotes the latent features of nodes, the Dirichlet energy is defined as:

$$E^{dir}(\mathbf{Z}^i) = \sum_{(u,v) \in \mathcal{E}^i} \left\| \mathbf{Z}_u^i / d_u - \mathbf{Z}_v^i / d_v \right\|_2^2 \tag{2}$$

where $\mathbf{Z}_u^i$ denotes the representation of node $u$. $E^{dir}(\mathbf{Z}^i)$ sums the squared differences of the feature vectors across all edges. Intuitively, $E^{\text{dir}}$ is small when the connected nodes have similar representations (smooth signal), and large when the neighboring nodes differ (indicating sharpening).

# 4 GNN ROBUSTNESS TO NOISY GRAPH LABELS, AND ITS FAILURES MODES

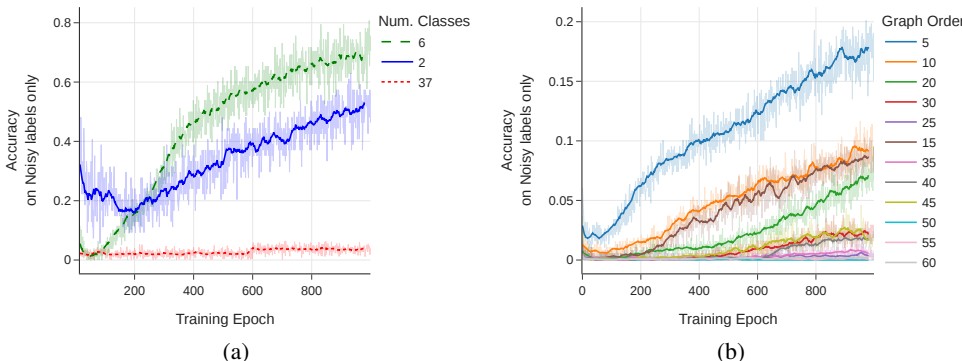

(a)       (b)

Figure 1: Training accuracy on noisy labels only. Effect of dataset properties: (a) Fewer classes in PPA lead to faster overfitting on noise. (b) Lower graph order leads to faster overfitting on noise..

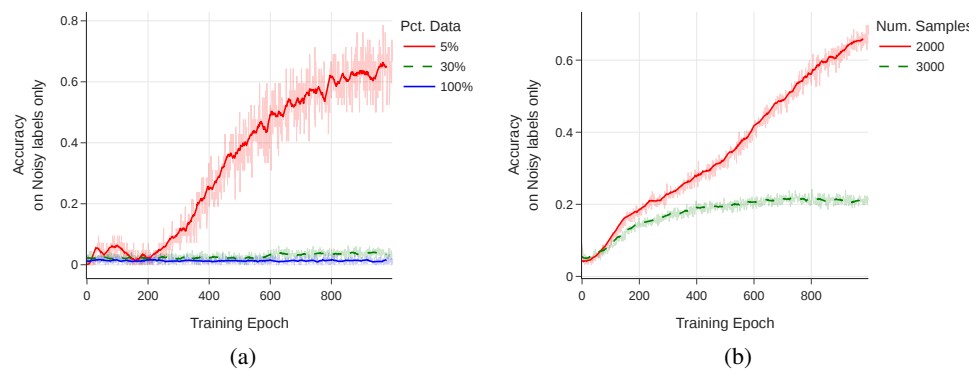

(a)                                                  (b)

Figure 2: Training accuracy on noisy labels only. Effect of dataset size: (a) Smaller fractions of the PPA dataset lead to stronger noise memorization. (b) Smaller synthetic datasets are also more prone to memorizing noise.

We study when and why a GNN overfits noisy labels. Interestingly, GNNs exhibit a degree of inherent robustness. For instance, injecting noise into the full PPA dataset Szklarczyk et al. (2018), or large portions of it, does not significantly degrade model performance (see Fig. 1(a), 2(a), 6). We hypothesize that the observed robustness on PPA stems from the fact that the models may be under parameterized for the inherent difficulty of the PPA benchmark, on which state of the art methods struggle to achieve perfect accuracy[1]. This finding aligns with general findings that under parameterized models are often more robust to noise (Zhang et al., 2017a). Despite this, we show that GNNs nevertheless fail under certain conditions of label noise. To examine this, we fix the model architecture and manipulate task complexity by (i) varying the number of classes as a proxy for over-parameterization, (ii) varying the average number of nodes in synthetic datasets, and (iii) varying the share of training data used. We study the average training accuracy on noisy labels as a direct measure of how much the model fits noise. Higher accuracy on noisy samples indicates the model is memorizing them, while lower accuracy suggests that the model is not fitting the noise.

**GNN Robustness varying number of classes** We use 30% of the PPA dataset and inject 20% symmetric noise into this subset by randomly replacing labels with uniformly sampled incorrect classes. The model here and below, if not said otherwise, is a 5-layer Graph Isomorphism Network (GIN) Xu et al. (2019) with 300 hidden units. Specifically, we create a sub-sampled PPA dataset with 2, 6, and the full 37 classes. Intuitively, reducing the number of classes simplifies the classification task and reduces the effective dataset size, making the fixed model increasingly overparameterized relative to the task. Fig. 1(a) shows the training accuracy on noisy samples across epochs. For the full 37-class task, the GNN does not memorize noise and remains relatively robust. However, when the number of classes is reduced to 6 and further to 2, the model increasingly fits the noisy labels. Interestingly, the 2-class case exhibits slightly more robustness than the 6-class case due to the symmetric nature of the injected noise, since random flipping between two classes produces highly contrasting noisy samples.

**GNNs are not robust on low-order graphs.** We generated synthetic datasets (see procedure in Appendix, Section B.2) to study the effect of graph order (number of nodes). As shown in Fig. 1(b), GNNs become increasingly sensitive to noise as the graph order decreases. Small graphs lack sufficient internal structure and aggregation capacity, making them vulnerable to treating noisy labels as signals. Conversely, larger graphs provide more nodes over which the model can average, diluting the influence of noisy samples.

**GNNs are not robust on small training sets.** The size of a training set affects the robustness to noisy labels. For the PPA dataset, we keep all 37 classes, but subsample the number of training graphs per class. As shown in Fig. 2(a), reducing the number of training samples increases the likelihood of overfitting noise. A similar trend is observed for the synthetic datasets (with graph order 7 and 6 classes) under 35% label noise, as shown in Fig. 2(b). In both cases, models trained on smaller

---

[1] https://paperswithcode.com/sota/graph-property-prediction-on-ogbg-ppa

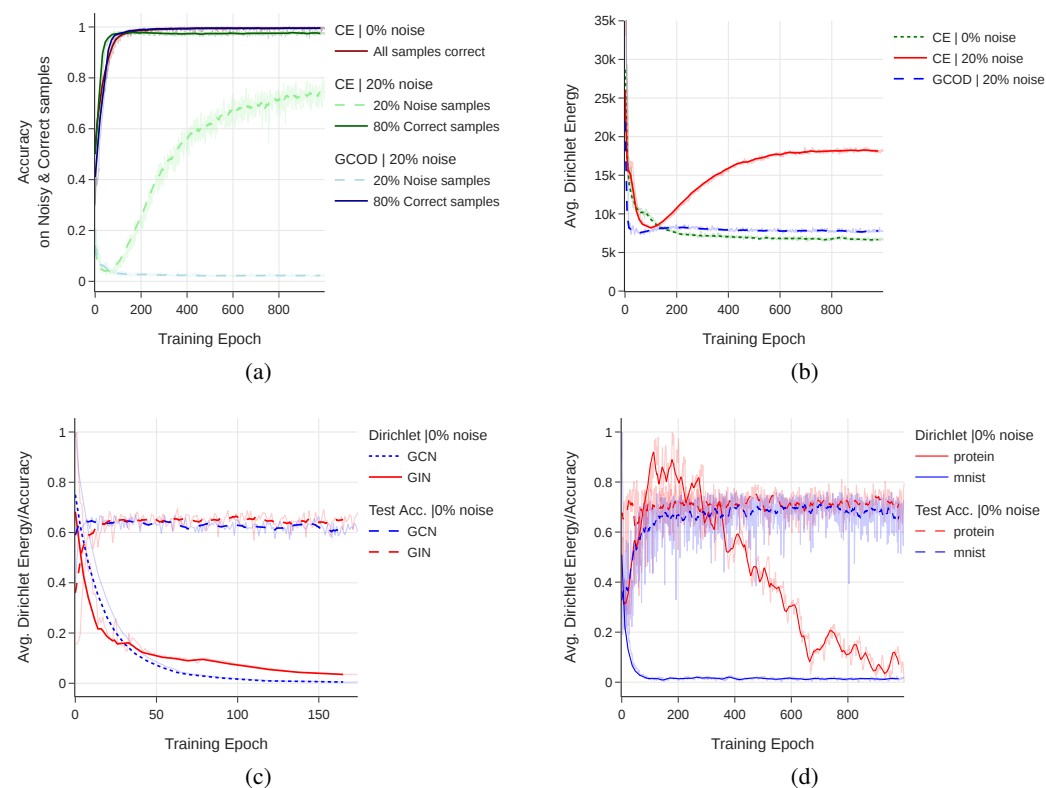

Figure 3: (a) Evolution of training Accuracy for GIN model on the PPA dataset (30% sample, 6 classes) with clean 0% or 20% label noise for CE and GCOD . (b) Dirichlet energy for clean and noise introduced PPA dataset (30% sample, 6 classes). The Dirichlet energy increases when the model with CE fits on noise. (c) Dirichlet energy and test Accuracy on the PPA dataset using CE with GIN and GCN models. (d) Dirichlet energy and test Accuracy on different datasets using GIN model (axes scaled for comparison).

datasets have a higher tendency to memorize noisy samples due to insufficient clean data to learn generalizable patterns.

## 5 REPRESENTATION DIRICHLET ENERGY INDICATES OVERFITTING ON NOISY LABELS

Across all experimental setups, we consistently observe that the Dirichlet energy ($E^{\mathrm{dir}}$) of the learned node representations increases once the model begins to fit on noisy labels (see Fig. 3(b)). During early training, when the model captures true patterns, $E_{\mathrm{dir}}$ remains low; however, as the model starts fitting the noisy samples, the energy increases significantly. This consistent behavior across diverse datasets and conditions ( Fig. 3(b) and Figure 4) motivates us to use $E^{\mathrm{dir}}$ as a signal to detect and monitor overfitting to label noise.

This leads to our first research question **RQ1**: *Is Dirichlet energy related to GNN performance on graph classification tasks, and how does it evolve when label noise is introduced during training?*

To address this, we propose a graph-level Dirichlet energy measure for the graph classification task and analyze its empirical behavior during training under noise conditions. Given a set of graphs $\mathcal{S}$, composed of graphs $\mathcal{G}^i$ with latent representation $\mathbf{Z}^i$, we define the Dirichlet energy of the set $\mathcal{S}$ as

$$E_{\mathrm{set}}^{dir}(\mathcal{S}) := \frac{1}{|\mathcal{S}|} \sum_{\mathcal{G}^i \in \mathcal{S}} E^{dir}(\mathbf{Z}^i). \tag{3}$$

In particular, we study $E_{\text{set}}^{dir}(\mathcal{D}_c)$, the average Dirichlet energy of graphs belonging to class $c$, where $\mathcal{D}_c$ denotes the subset of training graphs with the label $c$.

Our empirical analysis provides a consistent answer to **RQ1**. In clean datasets, i.e., without label noise, $E_{\text{set}}^{dir}(\mathcal{D}_c)$ may fluctuate during the initial training phase as the model begins to adjust to the task. However, we consistently observe a steady decrease in the later stages of training, culminating in low and stable Dirichlet energy values once the model converges to high classification accuracy. This behavior is illustrated in Fig. 3(c) and Fig. 3(d) for models trained with standard Cross-Entropy (CE) loss and no label noise (CE-0%).

However, when synthetic label noise is introduced (e.g., 20% symmetric flipping), the behavior diverges. As shown in Fig. 3(a), while the initial phase of training still exhibits a decrease in $E_{\text{set}}^{dir}(\mathcal{D}_c)$, this is followed by a significant increase during later epochs, precisely when the model starts to fit the noisy labels. This memorization phase is marked by rising training accuracy on noisy samples (see CE-20%), demonstrating a direct link between noise fitting and increased Dirichlet energy. This phenomenon is consistently observed across datasets and model architectures, including MUTAG, MNIST, and PROTEINS (see Fig. 3(d) and Fig. 9 in Appendix I). These findings confirm that Dirichlet energy serves as a reliable signal of representation smoothness and its disruption as a result of noise memorization.

Furthermore, to isolate the spectral dynamics, we utilize the HLFF-GNN framework Xu et al. (2024), which decomposes the node representations into low-frequency $\mathbf{Z}_1$ and high-frequency $\mathbf{Z}_2$ components. Experiments show that while $E^{dir}(\mathbf{Z}_1)$ remains stable, $E^{dir}(\mathbf{Z}_2)$ sharply increases during noise overfitting, confirming that high frequency energy components are responsible for fitting mislabeled data (detailed analysis of these experiments is provided in Appendix E).

From these observations, we conclude that maintaining a low Dirichlet energy, particularly by suppressing some high-frequency components, correlates with robust generalization. However, directly minimizing $E_{\text{set}}^{dir}(\mathcal{S})$ as a loss term presents practical challenges. First, the asymptotic energy level varies between datasets and architectures, making it difficult to define a universal target. Second, $E_{\text{set}}^{dir}$ is a global dataset level quantity, which is not easily decomposed into sample gradients for stochastic optimization. We propose alternative strategies to promote smoothness.

## 6 ROBUST STRATEGIES BASED ON SMOOTHING

### 6.1 METHOD 1: ROBUST GNN BY ENFORCING POSITIVE EIGENVALUES OF TRANSFORMATIONS

Our previous findings established a strong correlation between a GNN's overfitting of noisy labels and a significant increase in the Dirichlet energy of its learned node representations. The spectral properties of the learnable weight matrices within GNN layers fundamentally shape the network's behavior on the graph, particularly concerning smoothing and sharpening of features. Prior work Cai & Wang (2020); Oono & Suzuki (2021); Di Giovanni et al. (2023) has shown that the eigenvalues of learnable weight matrices interact with the graph Laplacian, inducing either smoothing or sharpening effects. In particular, Di Giovanni et al. (2023) demonstrates that positive eigenvalues promote attraction between connected nodes, while negative eigenvalues induce repulsion. These findings suggest that controlling the sign of the weight spectrum could architecturally enforce a smoothing inductive bias. This lead us to formulate our second research question: **RQ2** *Does the spectrum of the weight matrices affect the evolution of equation 3 during training?*

To justify our dataset level analysis of Dirichlet energy, we first present the following result:

**Proposition 6.1.** *Let $\mathcal{D} = \{\mathcal{G}^1 = (\mathbf{Z}^1, \mathbf{A}^1), \ldots, \mathcal{G}^n = (\mathbf{Z}^n, \mathbf{A}^n)\}$ be a set of graphs. Then $E_{set}^{dir}(\mathcal{D}) = \frac{1}{|\mathcal{D}|} E^{dir}(\mathbf{Z})$, where $\mathbf{Z} = [\mathbf{Z^1} \| \ldots \| \mathbf{Z^n}]$ and $\mathbf{A}$ is a block-diagonal matrix with blocks $\mathbf{A}^i$ along the diagonal. That is, the dataset-level Dirichlet energy corresponds to the Dirichlet energy of a single disconnected graph composed of all graphs in $\mathcal{D}$.*

**Remark 1.** *Reducing $E_{set}^{dir}(\mathcal{D})$ during training implies that the model is simultaneously enhancing low-frequency representations across all graphs in the dataset.*

Discussion of Proposition 6.1 and Remark 1 is provided in Appendix F.1. This result supports the idea that smoothing at the dataset level can be induced by controlling the local graph behavior. Motivated

by Di Giovanni et al. (2023), we hypothesize that smoothing can be enhanced by removing negative eigenvalues from the learned weight matrices of the GNN. To test this hypothesis, we constrain the spectrum of the weight matrix $\mathbf{W}^{(2)}$ in each GNN layer after neighborhood aggregation. We refer to this approach as **CE+W2**, which uses standard cross entropy loss but with post-hoc positive eigenvalue enforcement on $\mathbf{W}^{(2)}$. *The full derivation of the update rules, spectrum filtering, and implementation details are provided in Appendix C.1.* The performance of the method is reported in Table 1. Despite the findings affirm that controlling the weight matrix spectrum influences Dirichlet energy and robustness, the eigen decomposition step introduces severe training overhead (Table 7) and potential instability (Appendix Fig. 11).

## 6.2 METHOD 2: ROBUST GNN BY DIRECT ENERGY MANIPULATION

In this Section, We introduce a training method that explicitly constrains Dirichlet energy. By penalizing graphs with energy above a threshold, the model is encouraged to learn smoother, low-frequency representations, which we hypothesize improves robustness to label noise. Our approach is motivated by the empirical observations presented in Section 5 and Appendix G, which show that Dirichlet energy increases under overfitting and grows proportionally with label noise.

Formally, for a training set $\mathcal{D} = \{\mathcal{G}_1, \ldots, \mathcal{G}_N\}$ with associated Dirichlet energies $E_i = E^{dir}(\mathcal{G}_i)$ and class labels $c_i$, we define the regularization term:

$$\mathcal{L}_{DE}(\mathcal{D}) = \frac{1}{N} \sum_{i=1}^{N} \left[\max(0, E_i - U_{c_i})\right]^2, \tag{4}$$

where $U_{c_i} \in \mathbb{R}$ is the energy threshold for class $c_i$. The overall training loss becomes $\mathcal{L} = \mathcal{L}_{CE} + \lambda \mathcal{L}_{DE}$, where $\lambda$ balances the smoothness constraint against the classification objective. We explore two strategies for setting $U_c$ (see also Section C.4 for more details on these strategies):

**Class-specific bound:** A dynamic threshold $U_c$ computed after each epoch as the average Dirichlet energy of clean validation graphs in class $c$. The validation set must be clean to provide a reliable reference for estimating class-dependent energy levels. When noise is high, especially symmetric noise, the class-specific upper bounds $U_c$ may lose discriminative power as energy distributions across classes become similar, reducing the method's effectiveness. Using clean validation data preserves the class-specificity of the thresholds.

**Fixed bound:** A global threshold $U_c = U$ for all classes. In this case, the approach is not dynamic; instead $U$ is kept fixed during training and treated as a hyperparameter tuned to balance the need to prevent excessive smoothing while still limiting energy growth under noise.

On the PPA dataset with symmetric label noise, both variants of $\mathcal{L}_{DE}$ improved test accuracy over standard $\mathcal{L}_{CE}$. The fixed bound effectively constrained energy but occasionally over smoothed representations, particularly under clean labels. In contrast, class-specific bounds yielded better generalization and stability, improving accuracy under both noisy and clean conditions (Table 1). These results confirm that Dirichlet energy regularization helps stabilize feature evolution and enhances robustness by limiting harmful high frequency components.

## 7 METHOD 3: ROBUST GNN WITH GCOD LOSS FUNCTION

Having shown that noise overfitting aligns with increasing Dirichlet energy and that $E^{dir}$ decreasing methods improve robustness, we now explore an alternative path; a robust loss function. We introduce Graph Centroid Outlier Discounting GCOD , adapted from Wani et al. (2024) image classification for learning with noisy labels. Unlike previous methods, GCOD enhances robustness directly through its formulation, not by explicitly controlling $E^{dir}$ While GCOD is not designed to directly minimize Dirichlet energy, we investigate its performance in the presence of label noise and, crucially, observe the corresponding behavior in terms of $E^{dir}$. In this section, we focus on research questions: **RQ3**. *Is GCOD able to prevent learning of noisy samples and promote smoothness of equation 3, even though it is not specifically designed for it?* We analyze this question through a set of experiments.

NCOD Wani et al. (2024) is a loss function designed to address overfitting due to noisy labels for image classification (Zhang et al., 2017a). NCOD assumes samples of the same class are closer in latent space and leverages Deep Neural Networks' tendency to first learn from clean samples before noisy ones (Arpit et al., 2017).

Our GCOD method adapts the NCOD framework for graph classification with noisy labels. All the details about the newly designed loss are relegated to Section C.3 in the Appendix. We introduce two main modifications over the original NCOD. We add a third loss term $\mathcal{L}_3$ (see eq. equation 9 in Section C.3). This term uses a regularization based on a per sample trainable parameter to help the model distinguish between clean and noisy samples for better alignment. We incorporate the current training accuracy ($a_{train}$) as feedback into other loss terms (eq equation 7, 8 in Section C.3). This prioritizes learning from samples that the model correctly fits in early training, assuming these are more likely to be clean.

GCOD consistently reduces overfitting on noisy samples and preserves smoothness in graph learning, as evidenced by the decreasing Dirichlet energy (Fig. 3(a) and 3(b)). Our results for Graph Isomorphism Networks (GIN) Xu et al. (2019) in Table 1, Table 6, and (Fig. 10 and for Graph Convolutional Networks (GCN) Kipf & Welling (2016) in Fig. 7 in the Appendix) show that GCOD effectively mitigates noise impact, validating our hypothesis and answering **RQ3**.

Table 1: Performance on the PPA dataset using 30% of the data restricted to 6 selected classes. The best test accuracy is highlighted in bold red, the second best in blue. Reported values denote mean ± standard deviation across 4 independent runs

| Noise | Method | Test Acc. | | Train Acc. | |
|---|---|---|---|---|---|
| | | Best | Last | Best | Last |
| 0 % | CE | 96.25 ± 0.05 | 91.25 | 1.00 | 99.33 |
| | GCOD | 96.65 ± 0.52 | 93.25 | 99.23 | 99.03 |
| | CE + W2 | 96.50 ± 1.12 | 86.50 | 99.76 | 99.29 |
| | Fixed | 96.58 ± 0.27 | 85.70 | 99.26 | 98.29 |
| | Class-specific | 96.96 ± 0.04 | 88.67 | 99.41 | 98.67 |
| 20 % | CE (clean only) | 96.15 ± 0.09 | 90.66 | 84.50 | 84.07 |
| | CE | 88.66 ± 0.16 | 62.58 | 94.98 | 93.45 |
| | SOP | 91.01 ± 0.44 | 85.50 | 79.17 | 77.64 |
| | GCOD | **93.91 ± 0.26** | **92.58** | 80.11 | 79.52 |
| | CE + W2 | 89.83 ± 1.23 | 76.66 | 84.90 | 83.68 |
| | Fixed | 89.69 ± 0.57 | 80.25 | 78.07 | 70.14 |
| | Class-specific* | 92.34 ± 0.41 | 80.92 | 84.55 | 83.73 |
| 40 % | CE (clean only) | 95.08 ± 0.13 | 80.44 | 68.15 | 67.05 |
| | CE | 82.08 ± 0.22 | 51.83 | 82.47 | 78.88 |
| | SOP | 82.33 ± 1.32 | 65.41 | 58.90 | 57.68 |
| | GCOD | **93.88 ± 0.04** | **91.08** | 65.09 | 64.31 |
| | CE + W2 | 88.25 ± 1.13 | 56.33 | 68.78 | 65.88 |
| | Fixed | 88.58 ± 0.75 | 77.12 | 61.08 | 54.53 |
| | Class-specific* | 88.55 ± 0.11 | 71.83 | 68.98 | 67.80 |

* Requires clean validation set.

Table 2: Performance of the GIN network across multiple datasets under 40% asymmetric label noise. Reported values are test accuracy (%). Columns correspond to cross-entropy (CE) with clean labels (0% CE), cross entropy with noisy labels (40% CE), and GCOD under 40% noise (40% GCOD).

| Dataset | 0% CE | 40% CE | 40% GCOD |
|---|---|---|---|
| PROTEINS | 81.16 | 72.90 | 76.19 |
| MNIST | 72.69 | 71.15 | 72.61 |
| ENZYMES | 73.33 | 65.80 | 69.81 |
| IMDB/B | 76.50 | 68.18 | 72.89 |
| MUTAG | 94.73 | 91.16 | 93.19 |
| REDDIT | 48.15 | 48.01 | 47.94 |
| MSRC/21 | 96.69 | 94.39 | 95.57 |

Table 3: Percentage improvement over cross entropy (CE) using the GIN network. Values show gains of OMG and GCOD on selected datasets.

| Dataset | OMG | GCOD |
|---|---|---|
| MUTAG | 0.061 | 0.062 |
| IMDB-B | 0.047 | 0.049 |
| PROTEINS | 0.039 | 0.041 |

## 8 EXPERIMENTAL RESULTS

This section presents the empirical evaluation of our proposed GCOD loss function, along with the two methods leveraging Dirichlet energy regularization (CE+W2 and a method directly using $\mathcal{L}_{DE}$ with fixed and class-specific bounds), comparing their robustness against label noise across diverse datasets and conditions. In addition to standard Cross Entropy (CE) baselines, we include two recent state of the art methods for robustness: SOP Liu et al. (2022) and OMG (Yin et al., 2023). SOP is a leading approach for noise robust image classification based on sample reweighting, while OMG is a graph specific method designed to mitigate overfitting to noisy supervision. Their inclusion provides a strong comparative reference for evaluating the noise robustness of GCOD and our Dirichlet energy methods in both general and graph specific settings.

**Results on PPA** Table 1 summarizes the results under 0%, 20%, and 40% label noise on the PPA dataset, where 30% of the data across 6 classes was selected, utilizing a 5-layered GIN network (details provided in Appendix B)

The proposed GCOD loss consistently outperforms other methods by achieving a smaller gap between the best and final accuracies, which reflects improved generalization and robustness to noise. SOP, despite being competitive, exhibits wider accuracy gaps, indicating its susceptibility to overfitting. CE+W2 occasionally surpasses SOP at certain noise levels; its excessive smoothing leads to overfitting in noise-free scenarios. Precisely CE+W2 improves test accuracy under moderate noise (e.g., 20%: 89.83 vs. 88.66 compared to CE) and narrows the gap between training and test performance, indicating reduced overfitting. However, under clean labels (0% noise), CE+W2 tends to over-smooth, with slightly degraded final accuracy. The eigen decomposition step introduces a modest training overhead (see Table 7) and potential instability. However, the findings confirm that controlling the weight matrix spectrum influences Dirichlet energy and robustness. $\mathcal{L}_{DE}$ with a *Fixed bound* shows results in line with CE+W2: it's able to imporve the performance in the presence of moderate levels of noise, but in noise-free settings the gains were limited due to potential over-smoothing. With the use *Class-specific bounds*, instead, the adaptive mechanism allowed the regularization to align with the intrinsic complexity of each class, which enable the method to improve the accuracy at all levels of label noise, including the absence of noise. This suggests the class specific method improves generalization as well as model robustness.

**Results Across Multiple Datasets (20% Symmetric Noise)** Table 6 compares GCOD with standard CE under 20% symmetrical noise across several datasets. The Table shows that GCOD outperforms CE on most datasets, highlighting its resilience regardless of the specific data characteristics (with the exception of REDDIT-MULTI-12K in this specific test).

**Results under Asymmetric Noise (40%)** In Table 2, a comparison of GCOD and CE under 40% asymmetric label noise across datasets is presented. GCOD consistently outperforms CE, demonstrating its robustness in handling asymmetric label noise.

**Comparison with OMG** Lastly, Table 3 compares the percentage accuracy improvements of the OMG method and our proposed GCOD method across three datasets (MUTAG, IMDB-B, and PROTEINS) under experimental conditions similar to those in the OMG paper.It further underscores the superior performance and robustness of GCOD in noisy environments.

**Computational Efficiency and Hyperparameter Sensitivity of GCOD** Table 7 compares the percentage runtime increase for various methods relative to GIN trained with cross entropy loss, normalized to 1, CE+W2 incurs a 33% increase in training time. The GCOD loss function introduces no additional hyperparameters beyond the learning rate for the learnable parameters (the weights and a parameter $u$). Table 5 shows the impact of the learning rate of $u$ on GCOD performances.

# 9 CONCLUSIONS

In this paper, we examined GNN performance in graph classification under label noise. We identified robust scenarios where label noise has limited impact, but also highlighted GNN vulnerabilities where overly expressive models or low label coverage lead to performance drops. Unlike previous work, we explored robustness through an energy based lens, using the Dirichlet energy. Our findings show that learned smoother representations lead to better performance, while sharpness is linked to lower performance and noisy sample memorization. To make GNNs robust, we propose three methods: (i) inducing representation smoothness by relating graph Laplacian and weight matrix spectra; (ii) bounding the Dirichlet energy of representations in training; and (iii) offering GCOD loss function to enhance representation smoothness. All methods showed promising results without degrading performance in the absence of noise, which confirmed our hypotheses. **Limitations:** Although our work provides insight into the factors that influence GNN performance in the presence of noise, the experiments relied on limited theoretical foundations. In future work, we aim to theoretically explore the reasons leading to node feature sharpening in presence of noisy labels and investigate alternative applications of the Dirichlet energy in loss regularization. **Broader Impact and Outlook.** Beyond robustness to noisy labels, our findings suggest that Dirichlet energy may serve as a general lens for understanding the spectral dynamics of GNNs. We believe that controlling energy opens a path to principled design of architectures and losses that balance low and high frequency information, with potential benefits not only for noisy labels but also for other challenges.

## REPRODUCIBILITY STATEMENT

All implementation details and hyperparameter settings required to reproduce our results are provided in Appendix B. The source code is available at our GitHub repository `https://anonymous.4open.science/r/Robustness_Graph_Classification-E76F`.

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

# A  APPENDIX

This appendix provides additional details, extended experiments, and proofs that support the main paper. Section A describes the experimental settings and datasets. Section B presents further details on the proposed methods. Section C reviews additional related works. Section D–H contain extended experiments, ablation studies, and supplementary figures.

# B  EXPERIMENTAL SETTINGS

In our experimental setup, we performed tests on several datasets to evaluate performance. The first dataset, ogbg-ppa Szklarczyk et al. (2018), consists of undirected protein association neighborhoods derived from the protein-protein association networks of 1,581 species, spanning 37 broad taxonomic groups. Another dataset, ENZYMES Borgwardt et al. (2005), includes 600 protein tertiary structures from the BRENDA enzyme database, representing six different enzyme classes. The MSRC_21 dataset Neumann et al. (2016) contains 563 graphs across 20 categories, with an average of 77.52 nodes per graph. The PROTEINS Borgwardt et al. (2005) dataset is a binary classification set with 1,113 graphs, having an average node count of 39.06 per graph. The MUTAG Kriege & Mutzel (2012) dataset is another small binary graph dataset, consisting of 188 graphs, each with an average of 18 nodes. The IMDB-BINARY Yanardag & Vishwanathan (2015b) dataset, as the name suggests, is a binary graph classification dataset containing 1,000 graphs with an average node count of 20 per graph, and no node features. Similarly, the REDDIT-MULTI-12K dataset Yanardag & Vishwanathan (2015b) includes 11,929 graphs spread across 11 classes, with an average of 391 nodes per graph and no node features.

Additionally, we utilized the MNIST graph dataset, which is derived from the MNIST computer vision dataset. This dataset contains 55,000 images divided into 10 classes, where each image is represented as a graph.

Our experimental investigations were primarily conducted employing Graph Convolutional Networks (GCN) Kipf & Welling (2016) and Graph Isomorphism Networks (GIN) Xu et al. (2019) networks. Notably, the experimental methodology adopted possesses a generality that extends to encompass all Message Passing Neural Networks (MPNNs). Our study centers on observing the learning dynamics of networks during graph classification, particularly examining their adaptability to label noise. We aim to enhance robustness by employing tailored loss functions. Notably, the selection of hyperparameters remains unrestricted, as these parameters depend on both the model architecture and the dataset employed, ensuring a nuanced and generalized approach. In each experiment, we initialize with hyperparameters suited for clean, non-noisy conditions, ensuring optimal model performance. These parameters are subsequently held constant as we introduce varying levels of noise, sample density, or graph order. This approach ensures fair comparisons across experiments and facilitates a comprehensive exploration of model capacities. The synthetic label noise is generated following the methodologies described in Han et al. (2018b) and Xia et al. (2021) which are considered to be standard techniques for generating synthetic label noise.

## B.1  HYPERPARAMETERS

We employed the standard GIN Xu et al. (2019) and GCN Kipf & Welling (2016) architectures for most of our experiments. However, to investigate the impact of positive eigenvalues on weight matrices, as outlined in 6.1, we applied targeted modifications to both the GIN and GCN models C.1.

The table below summarizes the key hyperparameters used for the experiments.

All experiments have been performed over NVIDIA RTX A6000 GPU. For implementation, visit the following anonymous GitHub: Robustness Graph Classification Project.

## B.2  DETAILS ON THE SYNTHETIC DATASET USED FOR FIGURE 1(B)

Figure 1(b) presents results from synthetic datasets with varying graph order (i.e., number of nodes per graph). We generate datasets with average graph orders ranging from 5 to 60, sampling actual node counts from a Poisson distribution with mean equal to the target graph order. Each dataset

| Parameter | Value |
|---|---|
| Architecture | GCN, GIN |
| Learning Rate | 0.001 |
| Optimizer | Adam, |
| Batch Size | 32 |
| Loss Function | CrossEntropy, SOP and GCOD |
| Epochs | 200(PPA) to 1000 |
| Noise Percentage | 0% to 40% |
| Weight Decay | 1e-4 |
| Evaluation Metric | Accuracy |
| GNN Layers | 5 |
| Learning Rate $u$ | 1 |
| Hidden Units | 300 |

Table 4: Network architecture and hyperparameters.

contains 6 classes, with 1400 graphs per class, a fixed average degree of 2, and edges sampled uniformly at random.

Node features are sampled from a Gaussian distribution with a mean determined by the class label and a standard deviation of 1.5. For all graph orders, we apply a consistent label noise rate of 35% using uniform class flipping. Results demonstrate that GNNs become increasingly sensitive to noise as the graph order decreases. Small graphs lack sufficient internal structure and aggregation capacity, making them vulnerable to treating noisy labels as signal. Conversely, larger graphs provide more nodes and connectivity over which the model can average, diluting the influence of noisy samples.

## C    ADDITIONAL DETAILS ON THE THREE PROPOSED METHODS

### C.1    SPECTRAL BIAS IMPLEMENTATION DETAILS

#### C.1.1    GNN UPDATE RULES AND SPECTRAL ANALYSIS SETUP

To operationalize spectral bias in GNNs, we evaluate both GCN (Kipf & Welling, 2016) and GIN (Xu et al., 2019) layer update mechanisms. For each input graph $\mathcal{G}_i$, we define the GCN update rule as:

$$\text{GCN:} \quad \mathbf{H}_i^{l+1} = \sigma(\mathbf{\Delta}\mathbf{H}_i^l \mathbf{W}_l^1)\mathbf{W}_l^2, \quad \forall i \in \{1, \ldots, n\}, \tag{5}$$

and the GIN update rule as:

$$\text{GIN:} \quad \mathbf{H}_i^{l+1} = \sigma\left(\sigma\left((1+\epsilon)\mathbf{H}_i^l + \mathbf{A}_i\mathbf{H}_i^l\right)\mathbf{W}_l^1\right)\mathbf{W}_l^2,$$
$$\forall i \in \{1, \ldots, n\}. \tag{6}$$

Here, $\epsilon$ is a scalar hyperparameter and both $\mathbf{W}_l^1$ and $\mathbf{W}_l^2$ are square matrices of size $\mathbb{R}^{m' \times m'}$, allowing direct eigendecomposition. $\mathbf{W}_l^2$ is used as a shallow projection matrix after message aggregation.

### C.2    WEIGHT MATRIX SPECTRUM AND CLIPPING PROCEDURE

For each layer $l$ and each weight matrix $v \in \{1, 2\}$, let the eigenvalues and eigenvectors be denoted:

$$\{\mu_{0,l}^v, \ldots, \mu_{m'-1,l}^v\}, \quad \{\mathbf{\Phi}_{0,l}^v, \ldots, \mathbf{\Phi}_{m'-1,l}^v\}.$$

Unlike the graph Laplacian $\mathbf{\Delta}$, these eigenvalues $\mu_{i,l}^v$ can be negative, which enables feature "sharpening" effects. As shown in Di Giovanni et al. (2023), weight matrices with negative eigenvalues can amplify high frequency components often associated with noisy or irregular node signals.

To counteract this, we enforce a spectral bias by eliminating the influence of negative eigenvalues. The procedure is as follows:

1. Compute the eigendecomposition:

$$\mathbf{W}_l^v = \mathbf{\Phi}_l^v \boldsymbol{\mu}_l^v (\mathbf{\Phi}_l^v)^{-1},$$

where $\boldsymbol{\mu}_l^v$ is a diagonal matrix of eigenvalues.

2. Apply element-wise ReLU to retain only non-negative eigenvalues:

$$\boldsymbol{\mu}_l^{v+} = [\boldsymbol{\mu}_l^v]^+ = \max(\boldsymbol{\mu}_l^v, 0).$$

3. Reconstruct the filtered weight matrix:

$$\mathbf{W}_l^{v+} = \mathbf{\Phi}_l^v \boldsymbol{\mu}_l^{v+} (\mathbf{\Phi}_l^v)^{-1}.$$

This process removes sharpening components from the learned transformations, effectively biasing the GNN towards smooth solutions.

### C.2.1 TRAINING INTEGRATION AND BACKPROPAGATION HANDLING

In practice, we apply this spectral projection **after each gradient update**, treating it as a deterministic architectural constraint rather than part of the loss. The operation is not included in the computational graph—no gradients are propagated through the eigendecomposition or clipping.

This ensures that the model learns using unconstrained gradients, but the actual transformation used in forward passes remains positive-semidefinite.

### C.3 DETAIL ON THE DESIGN OF OUR GCOD

In our notation, $f_\theta : \mathbb{R}^{N \times m'} \to \mathbb{R}^{|C|}$ maps the final node representations $\mathbf{Z} \in \mathbb{R}^{N \times m'}$ to the class probabilities. We apply $f_\theta$ to batches of size $\mathbf{B}$, and introduce $\mathbf{u}_B \in \mathbb{R}^B$ as a trainable parameter, with $\hat{\mathbf{y}}_B$ as one-hot encoded class predictions, and $\tilde{\mathbf{y}}_B$ as calculated soft labels and as in Wani et al. (2024).

The $\mathbf{Z_B}$ is the tensor containing node representation for each graph in the batch, $\text{diag}_{\text{mat}}(M)$ Extracting the diagonal elements of a matrix $M$, while $\text{diag}_{\text{vec}}(\mathbf{v})$ construct a diagonal matrix from a vector $\mathbf{v}$. Here we offer its extension to Graph tasks with a new GCOD :

$$\mathcal{L}_1(\mathbf{u}_B, f_\theta(\mathbf{Z}_B), \mathbf{y}_B, \tilde{\mathbf{y}}_B, a_{\text{train}}) = \mathcal{L}_{\text{CE}}(f_\theta(\mathbf{Z}_B) + a_{\text{train}}\text{diag}_{\text{vec}}(\mathbf{u}_B) \cdot \mathbf{y}_B, \tilde{\mathbf{y}}_B), \tag{7}$$

$$\mathcal{L}_2(\mathbf{u}_B, \hat{\mathbf{y}}_B, \mathbf{y}_B) = \frac{1}{|C|} \left\| \hat{\mathbf{y}}_B + \text{diag}_{\text{vec}}(\mathbf{u}_B) \cdot \mathbf{y}_B - \mathbf{y}_B \right\|^2, \tag{8}$$

$$\mathcal{L}_3(\mathbf{u}_B, f_\theta(\mathbf{Z}_B), \mathbf{y}_B, a_{\text{train}}) = (1 - a_{\text{train}})\mathcal{D}_{KL} \left\{ \mathfrak{L}, \sigma\left(-\log\left(\mathbf{u}_B\right)\right) \right\} \tag{9}$$

where $\mathfrak{L}$ is $\log(\sigma\left(\text{diag}_{\text{mat}}(f_\theta(\mathbf{Z}_B)\mathbf{y_B}^T)\right))$ and $a_{\text{train}}$ is training accuracy.

Equation 9, is an additional term w.r.t vanilla NCOD, where we employ the Kullback-Liebler divergence as a regularization term to regulate the alignment of model predictions with the true class for clean samples (small $u$) while preventing alignment for noisy samples (large $u$). Moreover in equation 7, 8, we insert $a_{train}$ as a feedback term.

The parameters of the losses are updated using stochastic gradient descent as follows:

$$\theta^{t+1} \leftarrow \theta^t - \alpha\nabla_\theta(\mathcal{L}_1 + \mathcal{L}_3) \qquad \mathbf{u}^{t+1} \leftarrow \mathbf{u}^t - \beta\nabla_u\mathcal{L}_2 \tag{10}$$

The parameter $\mathbf{u}$ helps to reduce the importance of noisy labels during training, allowing the model to focus more on clean data. The computation of the soft label $\tilde{\mathbf{y}}_i \in \mathbb{R}^{|C|}$ (i.e. the $i$-th row of $\tilde{\mathbf{y}}$) relies on the concept of class embedding Wani et al. (2024).

### C.4 DETAILS ON THE DEFINITION OF THE BOUNDS FOR $\mathcal{L}_{DE}$

We defined two strategies for defining the upper bound for the regularization term $\mathcal{L}_{DE}$: a fixed global threshold and a class-dependent adaptive threshold.

**Class-dependent.** This approach is motivated by the observation that Dirichlet energy is influenced by factors that can be inherent to each class, such as graph topology. As a result, graphs from different classes may naturally exhibit distinct Dirichlet energy distributions.

To address this variability, as stated in the main text, we proposed the class-specific bound formulation. For each class $c$, the upper bound $U_c$ is computed at each epoch as the average Dirichlet energy over the clean validation graphs belonging to class $c$. Formally, given $\mathcal{D}_c^{val}$ the set of validation graphs in class c, and $E_i$ the Dirichlet energy of graph $\mathcal{G}_i$, the bound $U_c$ is computed as:

$$U_c = \frac{1}{|\mathcal{D}_c^{val}|} \sum_{\mathcal{G}_i \in \mathcal{D}_c^{val}} E_i \tag{11}$$

The use of clean validation data is essential for ensuring that the thresholds $U_c$ are reliable indicators of the intrinsic smoothness or complexity associated with each class. Relying on noisy samples to compute $U_c$, especially in the case of high symmetric label noise, would distort the energy, causing different class thresholds to collapse toward similar values. This would reduce the discriminative power of the regularization and lead $U_c$ to not be reflective of the true underlying structure of each class.
This adaptive strategy, then, ensures the regularization remains sensitive to plausible class-dependent variations between the distributions of the energy, which prevents the over penalization of inherently complex classes and under penalization of simpler ones.

**Fixed.** For the fixed settings, a constant threshold $U$ was applied uniformly across all the training samples, regardless of the class. This approach simplifies the regularization term and, by enforcing uniform penalization, provides a consistent regularization framework..

However, careful tuning of $U$ was necessary. If set too high, the regularization effect is negligible, allowing the model to overfit noise; if set too low, excessive smoothing occurs, causing a notable drop in accuracy due to the model's reduced ability to capture and distinguish important variations in the data. Consequently, $U$ was progressively decreased during experimentation until such a performance drop became evident. The selected value of $U$ thus represents a trade-off: energy is sufficiently reduced to prevent overfitting on noisy labels, while maintaining the model's capacity to distinguish between classes.

## D  ADDITIONAL RELATED WORKS

**Learning under label noise.** Some methods focus on sample relabelling (Arazo et al., 2019; Reed et al., 2014). Another family of techniques address noisy labels using two networks, splitting the training set and training two models for mutual assessment (Han et al., 2018a; Li et al., 2020; Kim et al., 2023). Regarding **regularization** for noisy labels, mixup augmentation Zhang et al. (2017b) is a widely used method that generates extra instances through linear interpolation between pairs of samples in both image and label spaces. Additionally, exist also **Reweighting techniques** aiming to improve the quality of training data by using adaptive weights in the loss for each sample (Liu & Tao, 2015; Pleiss et al., 2020).

**Graph Learning in noisy scenarios.** Works on node classification under label noise attempt to learn to predict the correct node label when a certain proportion of labels of the graph nodes are corrupted. In Du et al. (2023), authors exploit the pairwise interactions existing among nodes to regularize the classification. Other approaches use regularizes that detecting those nodes that are associated with the wrong information. Among these are contrastive losses Yuan et al. (2023a); Li et al. (2024), to mitigate the impact of a false supervised signal. Then in Yuan et al. (2023b), it was also proposed a self supervised learning method to produce pseudo labels assigned to each node. Other mechanisms that employ pseudo-labels are discussed in Qian et al. (2023), showing different policies to down weight the effect of noisy candidates into the final loss function.

The parallel line of work concerning GNN under noise is related to noise coming from missing or additional edges, and also noisy features. In Fox & Rajamanickam (2019), they focus on structural noise. They show that adding edges to the graph degrades the performance of the architecture. And propose a node augmentation strategy that repairs the performance degradation. However, this method is only tested with synthetic graphs. In Dai et al. (2022), they develop a robust GNN for both noisy

graphs and label sparsity issues (RS-GNN). Specifically, they simultaneously tackle the two issues by learning a link predictor that down weights noisy edges, so as to connect nodes with high similarity and facilitate the message passing. RS-GNN uses a link predictor instead of direct graph learning to save computational cost. The link predictor is MLP-based since edges can be corrupted. Their assumption is that node features of adjacent nodes will be similar. Once the dense adjacency matrix is reconstructed it is used to classify nodes through GCN. Even though these methods achieve state of the art performance they are specifically designed for node classification and have some assumptions on the input graph, such as the homophily property (Dai et al., 2022; Du et al., 2023; Yuan et al., 2023a; Dai et al., 2021). Moreover, some of these are validated only within graphs with the same semantics Dai et al. (2022); Du et al. (2023); Yuan et al. (2023b); Li et al. (2024); Dai et al. (2021) (e.g. citation networks), where the homophily assumption could be valid, but limiting for the overall research impact.

**Dirichlet Energy.** Graph Neural Networks (GNNs) face several challenges, including limited message passing expressiveness Morris et al. (2021), over smoothing Oono & Suzuki (2021), and over-squashing Alon & Yahav (2021). Over-smoothing has been studied using Dirichlet energy Zhou & Schölkopf (2005), which quantifies signal smoothness across graph nodes. Previous research explores the relationship between energy evolution and over smoothing Cai & Wang (2020); Nt & Maehara (2019), highlighting design choices that exacerbate this issue. Various approaches have been proposed to mitigate over-smoothing using energy properties Bo et al. (2021); Zhou et al. (2021a); Chen et al. (2023), though they are focused on node classification, where over-smoothing severely impacts performance (Yan et al., 2022). This narrow focus leaves unexplored how energy dynamics affects other graph tasks. In this work, we provide theoretical and practical insights on leveraging Dirichlet energy to enhance graph classification performance, even in the presence of label noise.

**Smoothing bias.** Most GNNs function as low-pass filters, emphasizing low-frequency components while diminishing high-frequency ones Nt & Maehara (2019); Rusch et al. (2023). Specifically, Nt & Maehara (2019) showed this phenomenon holds for graphs without non-trivial bipartite components, with self-loops further shrinking eigenvalues. Similarly, Topping et al. (2022) finds that non-bipartite graphs, especially without residual connections, exhibit low-frequency dominance. They also show that continuous-time models like CGNN, GRAND, and PDE-GCND maintain low-pass filtering. Cai & Wang (2020); Oono & Suzuki (2021) prove that GNN Dirichlet energy exponentially decreases with additional GCN layers when the product of the largest singular value of the weight matrix and the largest eigenvalue of the normalized Laplacian is less than one. Here, it is important to emphasize that Kang et al. (2018) examines graph classification under label noise using the mix-up technique. While the mix-up may indirectly promote smoothness in the graph, they do not discuss or establish a relationship between graph smoothness and the Dirichlet energy. Furthermore, their work centers on the smoothness of clusters within the graphs, rather than on the overall smoothness of the graph structure.

## D.1 Lipschitz Continuity in Graph Neural Networks

Regarding Lipschitz continuity, a key aspect of model robustness, Chuang & Jegelka (2022) provides a theoretical bound on the Lipschitz constant of the Graph Isomorphism Network (GIN) with respect to the Tree Mover's Distance (TMD). The derived bound, $|h(G_a) - h(G_b)| \leq \sum_{l=1}^{L+1} K_\phi^{(l)} \cdot \mathrm{TMD}_w^{L+1}(G_a, G_b)$, relates the change in the GIN's output to the distance between the input graphs as measured by TMD. This theorem highlights that if the constituent learnable functions $\phi^{(l)}$ have bounded Lipschitz constants $K_\phi^{(l)}$, then the entire GIN architecture exhibits a Lipschitz property with respect to TMD. Notably, TMD serves as a pseudometric for graphs that are distinguished by the $L$-iteration Weisfeiler-Leman (WL) test, a crucial property given that GIN's representational power is closely tied to the WL test. Davidson & Dym (2024) further contribute to the understanding of Lipschitz properties in neural networks operating on sets of features, which are fundamental building blocks in MPNNs. Their analysis of ReLU summation, a common aggregation function, demonstrates that it is uniformly Lipschitz under certain conditions. Moreover, their informal theorem on Hölder MPNN embeddings suggests that if the aggregation, combination, and readout functions within an MPNN are Lipschitz continuous, then the overall MPNN will also be Lipschitz continuous. Juvina et al. (2024) delve into tight Lipschitz constraints for GNNs in the context of node classification. By analyzing a generic graph convolution operation, they derive an optimal Lipschitz constant $\vartheta = \phi(\lambda_K)$ for the network, where $\lambda_K$ is the maximum eigenvalue of a weighted adjacency matrix

$M$, assuming non-negative weights and ReLU activations without bias. This work provides a more precise characterization of the robustness of GNNs to input perturbations. Gama et al. (2020) examine the stability of GNNs with respect to perturbations in the graph shift operator. Their Theorem 4 establishes that if the graph shift operator $S$ is perturbed by $E$ such that $|E| \leq \epsilon$, and the filter banks used in the GNN are bounded and the non-linearity is Lipschitz continuous, then the output of the GNN with the perturbed graph $\hat{S}$ will be close to the output with the original graph $S$, with a bound proportional to $\epsilon$ and the number of layers.

## D.2 Oversharpening in Graph Neural Networks

The primary definition of GNN oversharpening, as introduced in the literature and particularly highlighted by analyses such as Di Giovanni et al. (2023), characterizes it as an asymptotic behavior. Specifically, oversharpening occurs when the node features, after passing through multiple GNN layers, become predominantly determined by their projection onto the eigenvector of the graph Laplacian associated with its highest frequency. This implies that the learned representations capture primarily the most rapidly varying components of the signal over the graph. Pioneering work, notably by Di Giovanni et al., has rigorously established how the eigenvalues of GNN weight matrices directly influence feature dynamics, leading to either smoothing or sharpening effects. This analysis primarily considers linear graph convolutions employing symmetric weight matrices W. The key findings are: **Positive eigenvalues of W**: These induce an attractive force between the features of connected nodes. This attraction causes their representations to become more similar, promoting a smoothing effect across the graph. Consequently, features tend to align with the low-frequency components of the graph Laplacian, which is characteristic of oversmoothing. **Negative eigenvalues of W**: Conversely, these induce a repulsive force between the features of connected nodes. This repulsion drives their representations apart, leading to increased differences and thus a sharpening effect. This enhances the high-frequency components of the features. If these negative eigenvalues are sufficiently dominant and interact appropriately with the graph Laplacian's spectrum, this can lead to the oversharpening phenomenon, where node features become primarily aligned with the highest-frequency eigenvector of the graph Laplacian. The spectral norm of GNN weight matrices, while not a direct cause of oversharpening in the same way as the sign of eigenvalues, plays a significant modulatory role. It governs the overall "energy" or "scale" of the transformations applied by the GNN layers, thereby influencing the potential for various spectral phenomena, including oversharpening. The link between a large spectral norm (or large weight variance) and "oversharpening" (defined as high-frequency dominance) is indirect but significant. A large spectral norm, by definition, allows for eigenvalues of large magnitudes, both positive and negative. If learning dynamics or initialization conditions lead to a scenario in which negative eigenvalues of large magnitude become dominant within this expanded spectral envelope, the oversharpening conditions, as described by Di Giovanni et al. (2023), could be met.

Zhou et al. (2021b) analyzes this issue through the lens of Dirichlet energy, a measure of the variance of node embeddings. This work shows that the Dirichlet energy at each layer of a Graph Convolutional Network (GCN) is bounded by the Dirichlet energy of the previous layer, scaled by the singular values of the weight matrix. By imposing constraints on the Dirichlet energy, it is possible to control the smoothness of the learned embeddings. The work titled "Graph Neural Networks Do Not Always Oversmooth" challenges the universality of the oversmoothing problem. It establishes a "chaotic, non-oversmoothing phase" in GCNs that can be reached by appropriately tuning the weight variance at initialization. This suggests that oversmoothing is not an inherent limitation of GCN architectures, but rather a consequence of parameter initialization. Eldan et al. (2017)'s lemma on the spectral gap and edge addition provides insights into how graph structure influences spectral properties, which are related to information propagation and potentially oversmoothing. Their result shows that adding an edge can decrease the spectral gap of the Laplacian matrix under certain conditions related to the eigenvector and degrees of the connected nodes. Finally, the paper Zhuo et al. (2024) demonstrates that with carefully chosen weights, GNNs can avoid oversmoothing even in deep architectures. Specifically, by employing a whitening transformation on the node features at each layer, the network can prevent the convergence of node representations to a constant vector, suggesting that learnable weights play a crucial role in mitigating oversmoothing.

# E DIRICHLET ENERGY AMPLIFICATION IN HIGH-FREQUENCY COMPONENTS UNDER LABEL NOISE: A THEORETICAL AND EMPIRICAL ANALYSIS

In continuation of the findings presented in Section 5, where we established the role of Dirichlet Energy in identifying overfitting in noisy settings, we now deepen this perspective by dissecting the learned representations into frequency components. While we previously observed that elevated Dirichlet Energy in the later phases of training corresponds with the onset of noisy label fitting, our objective here is to uncover which representation subspaces are most impacted, and to understand the underlying dynamics. To this end, we utilize the HLFF-GNN framework Xu et al. (2024), "implemented in our work as `FGRLConv`",to demonstrate that high-frequency components bear the brunt of overfitting when GNNs are trained on noisy labels.

Empirical results already showed that the total Dirichlet Energy of graph representations tends to rise as the model begins fitting corrupted labels. However, this increase is not uniform across all representation spaces. The HLFF-GNN architecture offers a decomposition into three orthogonal signals: $Y$ (shared residual), $Z_1$ (low-frequency), and $Z_2$ (high-frequency). We hypothesize, and confirm, that it is the high frequency subspace $Z_2$ that is most vulnerable to label noise. This hypothesis, tested under graph classification (an extension beyond the original node classification setting of HLFF-GNN), is validated both theoretically and empirically.

Consider a graph $\mathcal{G} = (\mathcal{V}, \mathcal{E}, \mathbf{X})$, where $\mathcal{V}$ is the node set, $\mathcal{E} \subseteq \mathcal{V} \times \mathcal{V}$ the edge set, and $\mathbf{X} \in \mathbb{R}^{N \times m}$ the input feature matrix. Within HLFF-GNN, node features evolve through frequency modulated propagation into three latent subspaces (Xu et al., 2024). In our `FGRLConv` implementation:

- $Y$ represents the residual representation,
- $Z_1$ encodes low-frequency, smooth features propagated via message passing,
- $Z_2$ captures high-frequency, local features filtered using the graph Laplacian $\mathbf{\Delta}$.

These representations are updated as follows:

$$Y^{(l+1)} = P - \beta A_Z^{(l)}, \quad Z_1^{(l+1)} = Z_1^{(l)} - \frac{\beta}{\lambda} A_{YZ_1}^{(l)}, \quad Z_2^{(l+1)} = \mathbf{\Delta} Z_2^{(l)} - \frac{\beta}{\alpha} A_{YZ_2}^{(l)}$$

where $A_{YZ_1}, A_{YZ_2}, A_Z$ are batch aware attention mechanisms modulating signal interactions (Xu et al., 2024). The model minimizes a composite loss:

$$\mathcal{L} = \|Y - X\|_F^2 + \lambda \operatorname{tr}(Z_1^\top \mathbf{\Delta} Z_1) + \alpha \operatorname{tr}(Z_2^\top (I - \mathbf{\Delta}) Z_2) + \beta(\|Y^\top Z_1\|_F^2 + \|Y^\top Z_2\|_F^2)$$

Under clean labels, the objective guides the model toward smooth, interpretable feature spaces. However, in the presence of noisy supervision, the model is forced to encode erroneous patterns, disproportionately influencing $Z_2$. In the spectral domain, the Dirichlet Energy for $Z_2$ becomes:

$$E^{\mathrm{dir}}(Z_2) = \sum_{r=1}^{m} \sum_{u=0}^{N-1} \lambda_u (\psi_u^\top Z_{2r})^2$$

where $\lambda_u$ and $\psi_u$ are eigenvalues and eigenvectors of the Laplacian. Larger $\lambda_u$ correspond to higher frequencies, thereby exaggerating the effect of noise on the $Z_2$ energy profile.

To verify these dynamics, we trained `FGRLNet` on the ENZYMES dataset under both clean labels and 30% symmetric label noise. We tracked average per class per sample Dirichlet Energy for $Y, Z_1, Z_2$ across training epochs. Under clean supervision, $Z_2$'s energy increased modestly, while $Z_1$ and $Y$ either stabilized or declined. In contrast, noisy supervision triggered a sharp and continuous rise in $Z_2$'s energy, marking it as a reliable signal of overfitting. This phenomenon is visualized in 4.

Furthermore, statistical descriptors such as slope and standard deviation of $E_{\mathrm{dir}}(Z_2)$ were found to be strong early indicators of label noise. Under clean conditions, these metrics remained stable, but they deviated significantly under noisy labels, especially for mislabeled graphs.

These insights lead to actionable strategies for robust training:

- High-frequency components ($Z_2$) are principal amplifiers of label inconsistencies.

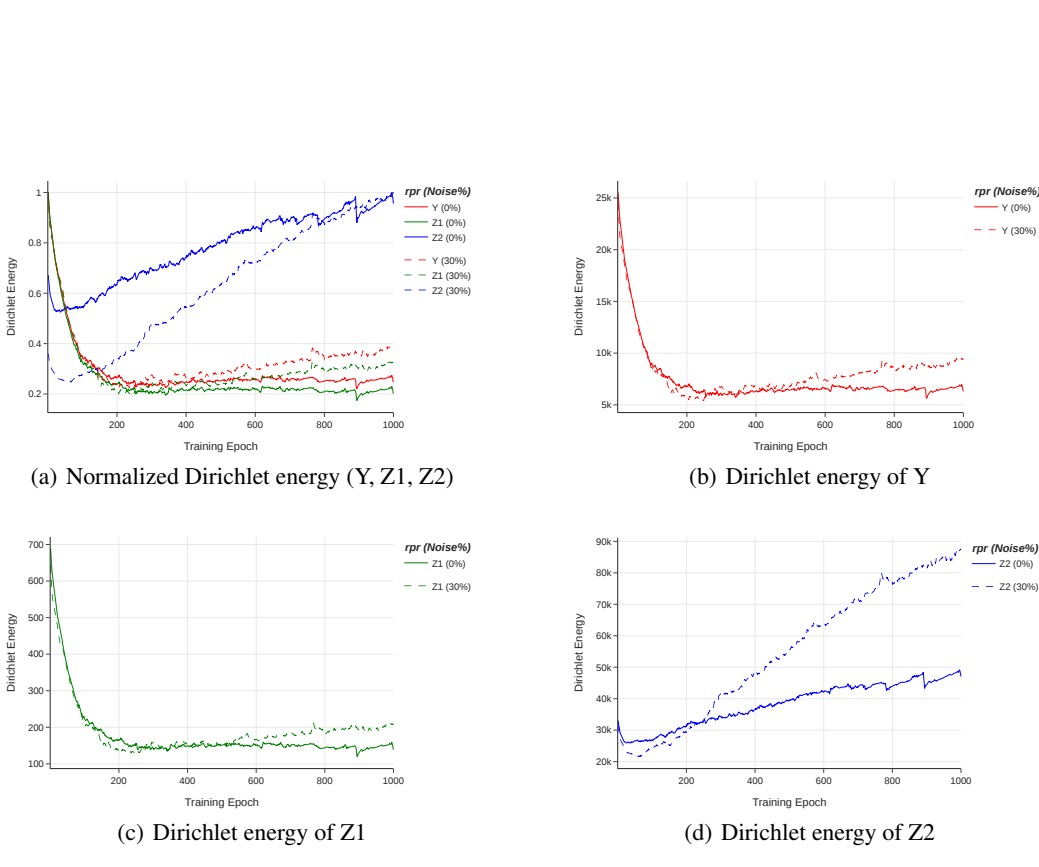

(a) Normalized Dirichlet energy (Y, Z1, Z2)

(b) Dirichlet energy of Y

(c) Dirichlet energy of Z1

(d) Dirichlet energy of Z2

Figure 4: Evolution of Dirichlet energy across training epochs for the representations $Y$, $Z_1$, and $Z_2$ learned by the FGRL model on the ENZYMES dataset. Solid lines represent training with 0% label noise, while dashed lines correspond to 30% symmetric label noise. The top-left plot shows the normalized energy trajectories for all three representations, with each normalized by its own maximum value to enable direct comparison. The remaining plots display the raw Dirichlet energy for each representation individually, preserving their respective scales to emphasize magnitude differences and noise sensitivity.

- Monitoring $E^{\text{dir}}(Z_2)$ dynamics allows early identification of noise driven instability.
- Losses can be adaptively modulated to suppress noisy gradient propagation through $Z_2$.

By grounding robustness in frequency sensitive learning signals, we offer a principled mechanism that can be used to detect and curb overfitting. This analysis extends and reinforces the findings of Section 5, charting a refined path forward in noise resilient GNN design.

# F ROBUST GNN BY ENFORCING POSITIVE EIGENVALUES OF TRANSFORMATIONS

## F.1 PROOF OF SECTION 6.1

*Proof of Proposition* 6.1. Let us denote $\Lambda = \{\lambda_u^i, 0 \leq u \leq N^i \wedge 1 \leq i \leq n\}$ as the set of the all graph frequencies in $\mathcal{D}$ and we rewrite it as $\Lambda = \{\lambda_k | 0 \leq k \leq N_{tot} \wedge N_{tot} = \sum_{i=0}^n N^i\}$. This formalization is agnostic to the specific graph in the dataset.
From this we can easily rewrite equation 12 as follows:

$$E^{dir}(\mathcal{D}) := \frac{1}{|\mathcal{D}|} \sum_{r=1}^{m} \sum_{u=1}^{N_{tot}} \lambda_u (\boldsymbol{\psi}_u^\top \mathbf{Z}_r^{tot})^2, \tag{12}$$

having $\mathbf{Z}^{tot} \in \mathbb{R}^{N_{tot} \times m'}$ and $\boldsymbol{\psi}_u \in \mathbb{R}^{N_{tot} \times 1}$.
Let us assume now the case of a graph $\mathcal{G} = (\mathbf{Z}, \mathbf{A})$, where $\mathbf{Z} = [\mathbf{Z^1} \| \dots \| \mathbf{Z^n}] \in \mathbb{R}^{N_{tot} \times m'}$, and $\mathbf{A} \in \mathbb{R}^{N_{tot} \times N_{tot}}$ is a diagonal block matrix, where each block $i$ in the diagonal is $\mathbf{A}^i$. The resulting $E^{dir}(\mathbf{Z})$ can be computed as:

$$E^{dir}(\mathbf{Z}) := \sum_{r=1}^{m} \sum_{u=1}^{N_{tot}} \lambda_u' (\boldsymbol{\psi}_u'^\top \mathbf{Z}_r^{tot})^2, \tag{13}$$

Let's notice that equation 13 differs from equation 12 in their set of eigenvalues and eigenvectors, and the scaling factor $|D|$.
Let us now define the graph Laplacian of $\mathcal{G}$ as $\boldsymbol{\Delta} = \mathbf{I_{N_{tot}}} - \mathbf{D}^{-\frac{1}{2}} \mathbf{A} \mathbf{D}^{-\frac{1}{2}}$. Being $\mathcal{G}$ composed by disconnected graphs we can write its Laplacian as the following diagonal block matrix:

$$\boldsymbol{\Delta} = \begin{bmatrix} \mathbf{I}_{N^1} - (\mathbf{D}^1)^{-\frac{1}{2}} \mathbf{A}^1 (\mathbf{D}^1)^{-\frac{1}{2}} & \cdots & 0 \\ \vdots & \ddots & \vdots \\ 0 & \cdots & \mathbf{I}_{N^n} - (\mathbf{D}^n)^{-\frac{1}{2}} \mathbf{A}^n (\mathbf{D}^n)^{-\frac{1}{2}} \end{bmatrix} = \begin{bmatrix} \boldsymbol{\Delta}^1 & \cdots & 0 \\ \vdots & \ddots & \vdots \\ 0 & \cdots & \boldsymbol{\Delta}^n \end{bmatrix} \tag{14}$$

From 14, we can evince that the eigenvalues set of eigenvalues $\Lambda'$ of $\boldsymbol{\Delta}$ corresponds to the union of the eigenvalues for each Laplacian of the disconnected graphs s.t. $\Lambda' = \{\Lambda^i | 1 \leq i \leq n\}$. This derives from the property that $det(\boldsymbol{\Delta} - \lambda \mathbf{I}_{N_{tot}}) = \prod_{i=1}^{n} det(\Delta^i - \lambda \mathbf{I}_{N^i})$ (Anton & Rorres, 2014). So this proves that $\lambda_u \equiv \lambda_u'$, $\forall u$ in Equations 12 and 13.

For the eigenvectors, suppose $\boldsymbol{\psi}_j^i$ is the $j$-th eigenvector of $\boldsymbol{\Delta}^i$ corresponding to eigenvalue $\lambda_j^i$ (e.g. $j \in \{0, \dots, N^i\}$). We construct the corresponding eigenvector of $\boldsymbol{\Delta}$ through the diagonal block matrix properties. Formally, the corresponding eigenvector $\boldsymbol{\psi}_u'$ of $\boldsymbol{\Delta}$ corresponding to $\lambda_j^i$ is given by:

$$\boldsymbol{\psi}_u' = \begin{bmatrix} 0 \\ \vdots \\ 0 \\ \boldsymbol{\psi}_j^i \\ 0 \\ \vdots \\ 0 \end{bmatrix} = \begin{bmatrix} 0 \\ \vdots \\ 0 \\ \boldsymbol{\psi}_u \\ 0 \\ \vdots \\ 0 \end{bmatrix}$$

This vector $\boldsymbol{\psi}_u'$ satisfies the eigenvector equation for $\boldsymbol{\Delta}$:

$$\boldsymbol{\Delta} \boldsymbol{\psi}_u' = \lambda_u \boldsymbol{\psi}_u' = \lambda_u' \boldsymbol{\psi}_u'$$

From this, it follows always that $\boldsymbol{\psi}_u^\top \mathbf{Z}_r^{tot} = \boldsymbol{\psi}_j^{i\top} \mathbf{Z}_r^i$, $\forall r$.
Thus, it follows that $E^{dir}(\mathcal{D}) = |\mathcal{D}| \cdot E^{dir}(\mathbf{Z})$.

## G ADDITIONAL EMPIRICAL STUDIES ON DIRICHLET ENERGY BEHAVIOUR

To gain deeper insight into the behavior of the Dirichlet energy during training, we conducted two additional experiments. These studies aim to clarify how energy evolves under different training dynamic, specifically in scenarios of overfitting and varying levels of label noise.

**Overfitting on clean data**

The first experiment investigates the evolution of Dirichlet energy when a model is intentionally overfitted to clean data. We trained a GIN model on the ENZYMES dataset without regularization and with the explicit goal of fitting the training data completely. As shown in Figure 5(a), the model successfully overfits the training set, as evidenced by the near-perfect training accuracy and the large gap between training and validation accuracy.

Notably, the Dirichlet energy consistently increases throughout the training process. This finding suggests that an upward trend in energy is not necessarily caused by label noise, but may instead be a general result of overfitting. In particular, the model's growing capacity to memorize fine-grained details may lead to less smooth and more fluctuating feature representations, reflected by higher Dirichlet energy.

**Training under varying levels of noise**

The second experiment investigates how the Dirichlet energy evolves during training when the dataset contains varying levels of label noise. A GIN model was trained on the PPA dataset under symmetric label noise at rates of $10\%, 20\%, 30\%$, and $40\%$. During training, we tracked the evolution of the Dirichlet energy according to the noise rate.

As illustrated in Figure 5(b), all noise levels exhibit a similar pattern in energy evolution: an initial decrease followed by a rise. This U-shaped trajectory suggests that the model initially learns generalizable low-frequency patterns, then begins to memorize label noise, resulting in less smooth node representations and thus higher Dirichlet energy.

Crucially, we observe that higher label noise levels consistently lead to higher final Dirichlet energy. The $40\%$ noise curve ends with the highest energy, while the $10\%$ noise setting maintains the lowest. This trend highlights a direct relationship between label noise and energy growth, further suggesting that Dirichlet energy can serve as an indicator of the extent to which the model is fitting noise.

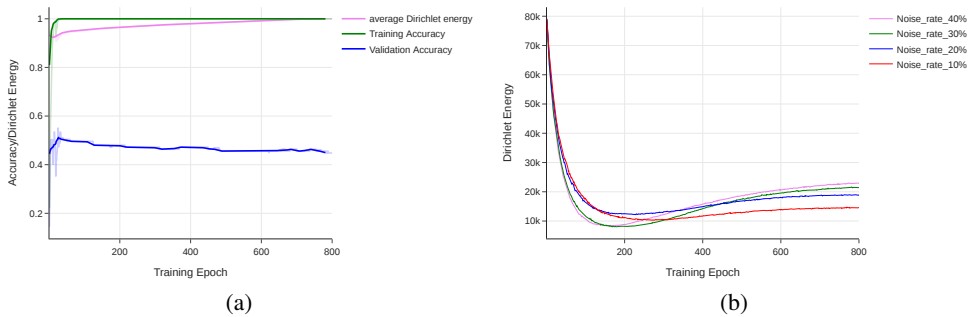

(a)             (b)

Figure 5: Empirical observations on Dirichlet energy dynamics. (a) Training on ENZYMES without noise, where the model is deliberately overfitted. The plot shows normalized Dirichlet energy alongside training and validation accuracy. As the model memorizes the training data, Dirichlet energy increases, indicating a rise in high-frequency components. (b) Evolution of normalized Dirichlet energy during training on PPA with symmetric label noise levels ($10\%$ to $40\%$). All curves follow a similar pattern: an initial energy decrease followed by a rise. Higher noise levels result in higher final energy, suggesting a link between Dirichlet energy growth and the amount of noisy labels.

## H ADDITIONAL TABLES

Table 5: Sensitivity analysis of $lr_u$ for $u$ under 40% asymmetric noise, %**Change** shows the percentage difference of test Accuracy using GCOD with two different learning rates for $u$.

| Dataset | $lr_u = 1$ | $lr_u = 0.1$ | % Change |
|---|---|---|---|
| Proteins | 76.19 | 75.89 | -0.39% |
| MNIST | 72.61 | 72.48 | -0.18% |
| Enzymes | 69.81 | 68.33 | -2.12% |
| IMDB/Binary | 72.89 | 71.94 | -1.30% |
| Mutag | 93.19 | 92.61 | -0.62% |
| Reddit | 47.94 | 48.08 | +0.29% |
| MSRC/21 | 95.57 | 95.45 | -0.13% |

Table 6: Performance of CE vs. GCOD with 20 % symmetric label noise. The last column reports the difference (GCOD – CE) on test accuracy.

| Dataset | Metric | 0 % CE | 20 % CE | 20 % GCOD | 20 % (GCOD–CE) |
|---|---|---|---|---|---|
| MNIST | Best | 72.69 | 66.30 | 71.26 | +4.96 |
|  | Last | 69.80 | 53.64 | 67.88 | +14.24 |
|  | Difference | 2.89 | 12.66 | 3.38 |  |
| ENZYMES | Best | 73.33 | 64.16 | 68.69 | +4.53 |
|  | Last | 65.78 | 57.50 | 62.54 | +5.04 |
|  | Difference | 7.55 | 6.66 | 6.15 |  |
| MSRC_21 | Best | 96.69 | 90.26 | 94.69 | +4.43 |
|  | Last | 93.80 | 79.64 | 90.26 | +10.62 |
|  | Difference | 2.89 | 10.62 | 4.43 |  |
| PROTEINS | Best | 81.16 | 76.23 | 79.38 | +3.15 |
|  | Last | 79.18 | 62.32 | 78.12 | +15.80 |
|  | Difference | 1.98 | 13.91 | 1.26 |  |
| MUTAG | Best | 94.73 | 89.47 | 90.01 | +0.54 |
|  | Last | 84.21 | 68.42 | 86.84 | +18.42 |
|  | Difference | 10.52 | 21.05 | 3.17 |  |
| IMDB-BINARY | Best | 76.50 | 75.00 | 75.40 | +0.40 |
|  | Last | 71.60 | 71.00 | 73.50 | +2.50 |
|  | Difference | 4.90 | 4.00 | 1.90 |  |
| REDDIT-MULTI-12K | Best | 48.15 | 45.05 | 44.98 | –0.07 |
|  | Last | 46.01 | 44.67 | 44.89 | +0.22 |
|  | Difference | 2.14 | 0.38 | 0.09 |  |

Table 7: Relative runtime comparison of GIN trained with standard Cross Entropy (baseline, runtime normalized to 1.00) versus alternative robustness-enhancing methods (SOP, GCOD, and CE+W2) on the PPA dataset (using 30% data, 6 classes).

| Method | Runtime |
|---|---|
| GIN | 1.00 |
| SOP | 1.048 |
| GCOD | 1.029 |
| CE+W2 | 1.33 |

# I  ADDITIONAL FIGURES

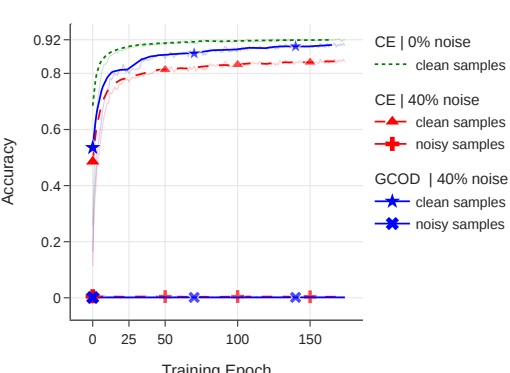

Figure 6: Training accuracy for known noisy and clean samples using GCN with CE loss. (4 class form PPA, with 40% symmetrical noise)

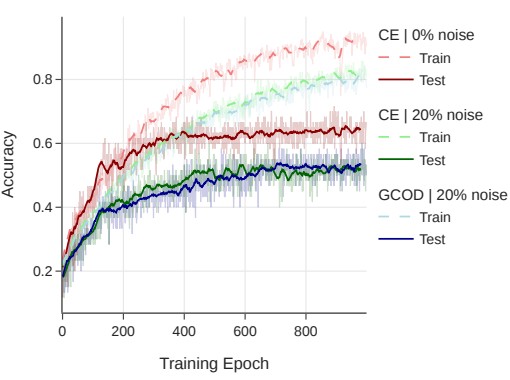

Figure 7: Comparison of the train and test accuracy for the Enzymes dataset. GCN model with different losses and noise levels.

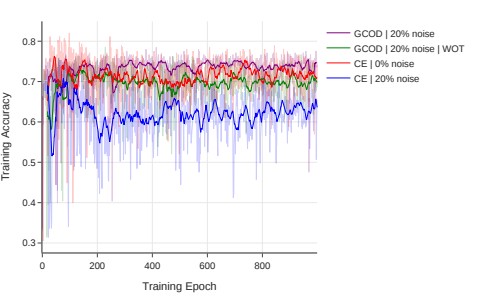

Figure 8: Ablation study showing that scaling $u$ by training accuracy prevents it from growing too aggressively. Without this scaling (GCOD WOT), $u$ dominates early and harms generalization, whereas in GCOD where $u$ is scaled by training accuracy activates $u$ gradually and achieves higher test accuracy.

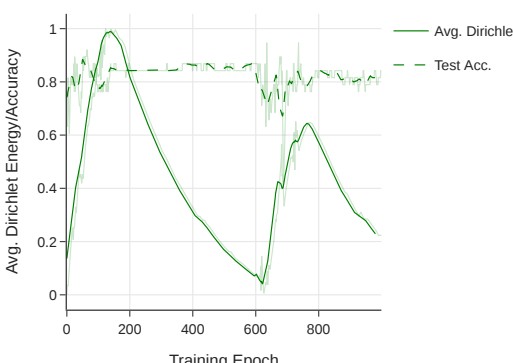

Figure 9: Average test accuracy and average Dirichlet energy on the MUTAG dataset with 0% label noise using the GIN model. The plot illustrates the evolution of accuracy and representation smoothness over training epochs.

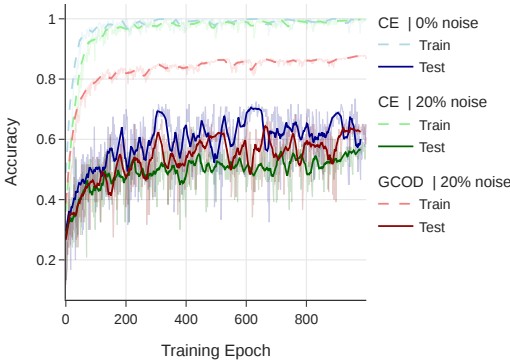

Figure 10: Comparison of the train and test accuracy for the Enzymes dataset with GIN model, on clean and 20% symmetric noise.

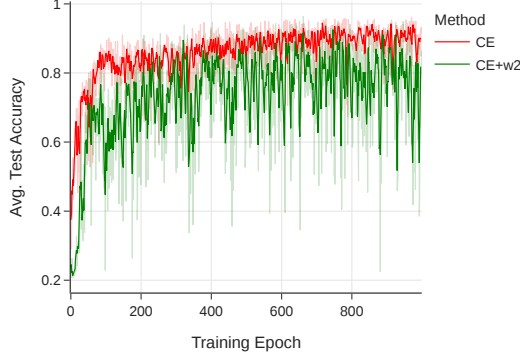

Figure 11: Test accuracy on the PPA dataset (30% subset, 6-class task) using Cross-Entropy (CE) and the CE+W2 method, which enforces positive-semidefinite weight matrices via eigendecomposition. While comparable peak accuracy, CE+W2 exhibits unstable convergence due to the spectra constraint applied after each epoch disrupts optimization.

