# OpenReview forum: "Energy Guided Smoothness to Improve Robustness"
_ICLR.cc/2026/Conference — Submitted to ICLR 2026_

### Official Review · Reviewer_TcSw · 2025-10-30

**Soundness:** 4
**Presentation:** 3
**Contribution:** 3
**Rating:** 8
**Confidence:** 4

**Summary:**

The paper examines the robustness of GNNs to label noise in the graph classification task. They empirically demonstrate that there is a correlation between the increase of Dirichlet energy when overfitted to noisy labels. Based on this finding, the paper proposed three training strategies: enforcing positive eigenvalues to focus on low-pass signals, explicit regularization of the Dirichlet energy, and a noise-robust loss function tailored for GNNs. Experimental results by comparing distinct noise levels further strengthen the contribution of this work.

**Strengths:**

- The paper tackles the challenge of noisy labels in graph classification, a problem that has received relatively limited attention compared to node classification.
- Through extensive empirical analyses in Section 4, the authors provide insights into the conditions under which GNNs fail to maintain robustness, and in Section 5, they establish a novel connection between this phenomenon and the Dirichlet Energy, demonstrating notable originality and potential for future research.
- Comprehensive experiments under both symmetric and asymmetric noise settings further substantiate the effectiveness and robustness of the proposed approach.

**Weaknesses:**

- The adopted GNN model primarily focuses on capturing low-pass signals. It would be beneficial to include a broader range of GNN architectures that can capture both low- and high-pass components, such as [1] and [2], to validate the generality of the findings.
- Figures 1 and 2 present results only under the noisy-label setting. If similar accuracy improvements are observed on clean graphs, the interpretation and discussion in Section 4 may require reconsideration.
- The claim that noise predominantly corresponds to high-frequency signals could be further supported through additional references or empirical evidence, which would help reinforce the justification of Method 1.
- Table 3 shows minor improvements compared to existing works.

Minor errors:
- Some expressions in the paper (maybe because of the usage of \eqref) should be modified (line 373, 849, and so on).
- The abbreviation GCOD should be explicitly defined upon its first appearance.
- Inconsistent expression in line 267 and equation 3, $\mathcal{G}_i$ and $\mathcal{G}^i$.


[1] GREAD: Graph Neural Reaction-Diffusion Networks, ICML 2023
[2] From Trainable Negative Depth to Edge Heterophily in Graphs, Neurips 2023

**Questions:**

1. Could the authors provide a more detailed explanation of how symmetric and asymmetric noise are defined, including the specific noise injection procedure used in the experiments?
2. In case of utilizing class-specific bounds for method 2, does the number of graphs for each class in the validation set affect the overall performance?
3. Rather than tuning hyperparameters on a clean graph, how would the performance trends change if the optimal hyperparameters were selected based on the noisy graph?
4. In Table 2, do other datasets exhibit a similar performance pattern between CE + W2 and the fixed, class-specific method as observed in the PPA dataset?

**Details Of Ethics Concerns:**

No Ethics Concerns.

---

> ### Author Response · Authors · 2025-11-20
>
> ### Weaknessess:
> 1.  This is a very insightful point, recent analyses of GNN inductive biases show that architectures naturally emphasize different regions of the graph frequency spectrum i,e, “different DGNs focus on different regions of the graph frequency spectrum, high-frequency DGNs rely on lower-order connectivity, while low-frequency DGNs exploit higher-order structure” [3]. Crucially, this frequency selectivity is an architectural property, and the notion of “high” or “low” frequency is always relative to the dataset’s Laplacian spectrum. When the data distribution shifts, the model’s operational band shifts with it. Because the Dirichlet energy view is based on relative frequency, our interpretation remains valid across low-pass, high-pass, and mixed-frequency GNNs, and GCOD remains architecture agnostic; it adapts to the energy profile observed during training. Although, we agree to test it emprically on other GNN architectures can be intresting.
>
>
> 2.   Figures 1 and 2 are designed specifically to analyze model behavior in the presence of label noise; under clean labels these figures cannot be meaningfully reproduced, since there are no noisy samples and the noise induced Dirichlet-energy effects do not arise. In the clean setting, all methods, including **GCOD** show similar accuracy, and **GCOD** provides little to no additional gain, which is expected because its correction term activates only when the model begins to fit noise. This supports the interpretation in Section 4: GCOD modifies the dynamics only when noise is present, not on clean graphs.
>
> 3.   A substantial body of work supports the fact that GNN acts as Low pass filters and perturbations in graphs act as  high frequency in graph representations [1]. In graph signal processing, high-frequency Laplacian components correspond to rapid variation across adjacent nodes [2]. Our own decomposition analysis in **Appendix E**  demonstrates that under label noise, the **Z2** high frequency subspace absorbs most of the amplified Dirichlet energy . This aligns with the recent frequency spectrum study of GNN inductive biases by [3], which shows that DGNs treat residuals as energy in the higher end of the Laplacian spectrum . Together, these works and our empirical results **(Figures 1–3, Appendix E)**   show that noise fitting corresponds to a rise in high frequency Dirichlet energy, reinforcing the justification behind **Method 1**.
>
> 4.  Please see our detailed response to Reviewer **WGPK**,  **Weakness 1**, which addresses this point directly.
>
> ### Minor issues: We thank you for pointing out these  issues. All of them have been corrected in the updated PDF included with this rebuttal.
>
>
> ### Questions:
> 1.  Please see our deatiled response to Reviewer **5D5b**, **weakness 1** which addresses this point directly
> 2.  To the class-specific bounds method, we noted that its reliability depends on the size of the clean validation set. On OGB-PPA with 40% noise, reducing the validation set to 30 or 10 samples per class drops accuracy from 88.55% to 79.33% and 67.00%, respectively. This sensitivity is precisely why we highlight GCOD as the main practical method, GCOD does not require clean labels or validation statistics and remains stable regardless of validation set size.
> 3.  **GCOD**  was designed to be hyperparameter light (only learning rate for **$u$**) and stable across a broad range of settings. The hyperparameters are always selected without the knowledge of the noise (doesn't need clean dataset), and it shows consistent improvements over CE, whereas methods relying heavily on clean validation signals (e.g., class specific bounds) may degrade the performance if we tune on noisy validation sets.
>
> 4. Yes, the pattern is consistent across datasets. Across PROTEINS, MNIST-Graph, ENZYMES, IMDB-B, and MUTAG, we observe the same trend:
>
>       * **CE + W2** shows occasional gains but suffers from instability due to per epoch eigendecomposition  and often underperforms GCOD.
>       * Fixed and class-specific bounds improve over CE but remain less robust than GCOD.
>
>     The PPA dataset highlights this clearly because it is large and structurally diverse, but the same qualitative behavior appears on smaller datasets as well (see Table 2 and Appendix, Section G  (Tables 6–7), Section I ADDITIONAL FIGURES).
> ---
> ### Refrences:
> 1.  A Survey on Oversmoothing in Graph Neural Networks $\textit{T. Konstantin Rusch, Michael M. Bronstein, Siddhartha Mishra}$
> 2.  The Emerging Field of Signal Processing on Graphs: Extending High-Dimensional Data Analysis to Networks and Other Irregular Domains $\textit{David I Shuman, Sunil K. Narang, Pascal Frossard, Antonio Ortega, Pierre Vandergheynst }$
> 3.  Bridging XAI and spectral analysis to investigate the inductive biases of deep graph networks $\textit{Michele Fontanesi,  Alessio Micheli,  Marco Podda,  Domenico Tortorella}$

---

### Official Review · Reviewer_WGPK · 2025-11-01

**Soundness:** 3
**Presentation:** 1
**Contribution:** 2
**Rating:** 4
**Confidence:** 4

**Summary:**

The paper presents an interesting connection between noisy label (model robustness) and Dirichlet energy under graph classification task. Specifically, from experiments, the author observes that models overfitting to noisy labels tend to have high Dirichlet energy rising and decreasing Dirichlet energy could provide more robust graph classification model. The author attemps with three different strategies to improve robustness of models under noisy labels by constraining learnable weight to positive eigenvalue only, adding Dirichlet energy as part of smoothness regularization loss, and a method inspired from image classification called GCOD loss. It is observed that despite not directly related to decreasing Dirichlet energy, GCOD loss still provide robust model and decrease Dirichlet energy.

**Strengths:**

1.The paper shows a clear observation between the connection of Dirichlet energy and label noise.
2.The paper provides comprehensive experiments on when models are overfitting to the noisy data and conclude with high dirichlet energy observation and propose three methods to alleviate it.

**Weaknesses:**

1. The proposed method show no clear difference between existing methods such as OMG as shown in Table 3.
2. The GCOD method description is too sparse in the main text, lacking detailed descriptions.
3. Why detailed results are only reported for PPA dataset with other baselines?

**Questions:**

1. I hate to ask but I have to ask again: why is the figure 3 legend still has the same typo? Is this something that is hard to change even after someone explicitly points it out?
2. See weakness.

---

> ### Author Response · Authors · 2025-11-20
>
> ### Weaknessess:
>
> 1.   Thank you for the comment and we would like to clarify that Table 3 reports **percentage improvements**, not raw accuracy deltas, which can visually compress the perceived gain even when the absolute robustness benefit is meaningful. More importantly, the key distinction is in **complexity**: OMG requires augmented graph views, dual forward passes, a reconstruction guided mixing network, and consistency regularization across views. In contrast, **GCOD is a pure loss level correction**, it introduces no architectural changes, no augmented views, and no additional modules, using only a single scalar update per sample.
>
>       If desired, GCOD can be combined with augmented views and KL divergence (as in OMG, NCOD+, JoCoR), since GCOD’s formulation is **orthogonal to the architecture**. Adding a second view simply adds one KL term to the GCOD loss without altering the method itself, and would further improve accuracy. The fact that GCOD already outperforms OMG without using any augmentations or dual view consistency demonstrates the inherent effectiveness of our approach.
>
>      In terms of **efficiency**, Table IV of the OMG paper reports an average runtime overhead of **~200%** for small datasets (MUTAG, PROTEINS, IMDB-B, NCI1) compared to a base GIN. OMG is even slower than Co-Teaching under identical settings. In contrast, GCOD has only a **0.029% overhead** even on the large and complex OGB-PPA dataset. This difference is expected since GCOD adds only a lightweight analytical correction, whereas OMG requires multiple stochastic augmentations, dual inference passes, and consistency filtering. Overall, even when the numerical improvements appear similar, GCOD provides a **far superior robustness-complexity trade-off**, outperforming OMG while being orders of magnitude simpler and more efficient.
>
>
>
>
> 2.   We agree that **GCOD is central to the paper**. Due to strict page limits, we moved the full description, optimization steps, and implementation details to the **Appendix**, where the method is described thoroughly. We may improve the explanation in the supplementary if it doesn’t satisfy currently.
>
>
>
> 3.   **PPA** is a large and challenging real world graph classification dataset, widely used in the GNN literature, making it ideal for a deeper analysis of robustness under scale and structural heterogeneity. For OMG specifically, we compare only on the datasets they originally evaluated on to ensure a **fair comparison** using identical settings. Because OMG did not evaluate PPA, we instead compare GCOD against CE, SOP, and other baselines on PPA. This design ensures fairness when comparing to OMG and completeness when demonstrating GCOD’s performance on larger, more complex datasets.
>
>
>
> ### Questions:
>
> 1. We sincerely thank you for catching this again. This was a **completely unintentional oversight**, not a technical difficulty. We apologize for the mistake and we **corrected it in the revised submission we uploaded**. We appreciate your patience.

---

### Official Review · Reviewer_5D5b · 2025-11-01

**Soundness:** 3
**Presentation:** 3
**Contribution:** 3
**Rating:** 6
**Confidence:** 3

**Summary:**

This paper focuses on the problem of overfitting and generalization degradation in graph classification tasks with noisy labels. The authors propose using Dirichlet energy as an observable signal to assess the degree of overfitting in GNNs and design three complementary strategies to enhance robustness. Experiments show that these methods significantly improve classification accuracy under label noise on several datasets, including PPA.

**Strengths:**

- The paper is well-written and easy to follow.

- The insights derived from the pre-experiments are compelling and directly motivate the three proposed methods.

**Weaknesses:**

- Both the pre- and main experiments rely on synthetic symmetric/asymmetric noise, rather than real-world noisy labels, limiting the external validity and generalizability of the conclusions.

- Dirichlet energy is a global measure for the whole dataset, while the proposed methods optimize at the sample level. This mismatch raises questions about scalability and effectiveness when applied to larger, more complex real-world graphs.

- The GCOD loss depends on training accuracy as feedback, which may introduce bias or instability during training. The paper does not thoroughly investigate or ablate this potential issue.

**Questions:**

Please see the weakness.

---

> ### Author Response · Authors · 2025-11-20
>
> ###  Weaknesses
>
> 1. Thank you for this comment, we want to emphasize that our synthetic noise protocols, both symmetric and asymmetric, are deliberately designed to reflect realistic annotation errors in graph-level classification. For symmetric noise, we follow the standard and widely used protocol in the noisy label literature, i,e, with probability $\eta$, a label $y$ is replaced uniformly by any other class $y' \neq y$, as done in NCOD [2], SOP[3], Co-Teaching [5], DivideMix[4], and JoCoR [1] . This provides a clean, controlled baseline where all classes are equally affected.
>
>      Moreover, our asymmetric noise is not random; we first train on clean labels, compute class centroids, measure inter class distances, and flip labels only toward the closest (most confusable) classes. This produces realistic failure modes e.g., molecular classes with similar functional groups, superpixel graphs with similar shapes, or social-network ego graphs with overlapping structures, and better reflects how real mislabeling occurs. Regarding real-world noisy graph datasets, we note that publicly available graph-classification benchmarks rarely contain verified label noise. To the best of our knowledge, no established graph-classification dataset provides ground truth noisy labels, unlike image benchmarks such as Clothing1M. For this reason, our noise design aims to approximate realistic annotation errors as faithfully as possible. We would be happy to evaluate GCOD on any real-world noisy graph dataset recommended by the reviewer.
>
>
>
>
>
>
>
>
> 2.  First we would like to know if this concern pertains specifically to **Method 2: ROBUST GNN BY DIRECT ENERGY MANIPULATION** as we imagine. We clarify below why there is no mismatch and how we address scalability and practical usage:
>
>      #### Connection between global and local energies:
>      There is no contradiction between computing Dirichlet energy globally and optimizing at the sample level. In fact, Dirichlet energy is strictly additive across graphs that said graph-level energies sum over a batch to approximate the dataset-level energy. This is formally established in **Proposition 6.1** of our paper, the dataset-level Dirichlet energy is equivalent to the energy of a disconnected super-graph composed of all samples. Therefore, the global Dirichlet energy is simply the aggregated form of the same local energies that govern representation smoothness during training, and the method remains fully compatible with mini-batch training.
>
>      #### Scalability and efficiency:
>     All computations required for Dirichlet energy (both local and aggregated) rely only on sparse Laplacian operations, which are already employed in standard GNN message passing (e.g., aggregation). As a result, the time and memory complexity scales naturally with the underlying GNN architecture. Empirically, we did not observe any scalability issues across datasets of diverse sizes, including large real-world benchmarks like **ogbg-PPA**.
>
>      #### Practical relevance and scope:
>      Importantly, our main contribution for practical use is **GCOD**, a noise-robust loss function which does not require computing validation energy or using clean labels. GCOD operates solely at the per-sample level and is effective in noisy environments without needing global Dirichlet energy monitoring. The Dirichlet-based method ($Method  2$) is included primarily to provide insights into the spectral underpinnings of noise overfitting and to validate our hypothesis, not as a dependency for GCOD.
>
>
>
>
>
> 3. In graph classification, learning progresses more slowly than in other domains due to irregular structure, weak inductive biases, and high intra class variability. For this reason, GCOD uses training accuracy as a lightweight and reliable proxy for model confidence, when accuracy is low, the correction term remains inactive to avoid suppressing useful gradients; as training accuracy improves, the regularization is gradually introduced, stabilizing training under noise. We did run the ablation ($ \text {APPENDIX Section I ADDITIONAL FIGURES Figure 8}$)  removing the accuracy feedback and applying a constant correction significantly reduced stability and degraded robustness. These observations confirm that a controlled ramp-up is essential for energy based regularization on graphs to allow the model to learn on earlier meaningful gradients more as compared to the later ones. We also want to emphasize  that the effect of this scaling is more predominent in bigger and complex graph datasets.
> ---
>
> #### References
>
> 1. Combating noisy labels by agreement: A joint training method with co-regularization
>
> 2. Learning with Noisy Labels through Learnable Weighting and Centroid Similarity
>
> 3. Robust Training under Label Noise by Over-parameterization
>
> 4. DIVIDEMIX: LEARNING WITH NOISY LABELS AS SEMI-SUPERVISED LEARNING
>
> 5. Co-teaching: Robust Training of Deep Neural Networks with Extremely Noisy Labels

---

> > ### Comment · Reviewer_5D5b · 2025-11-27
> >
> > Thank you for your response. It has clarified most of my doubts.
> >
> > However, I am curious if community-recognized benchmarks like NoisyGL[1] would be a valid alternative in the absence of real-world graph data. I assume using them would be more persuasive than manually synthesizing noise.
> >
> > Additionally, I would like to point out that the reference format in your rebuttal is not entirely standard, though this is a minor issue.
> >
> > [1] Wang, Z., Sun, D., Zhou, S., Wang, H., Fan, J., Huang, L. and Bu, J., 2024. Noisygl: A comprehensive benchmark for graph neural networks under label noise. Advances in Neural Information Processing Systems, 37, pp.38142-38170.

---

> > > ### Author Response · Authors · 2025-11-27
> > >
> > > Thank you so much for the follow up question.
> > >
> > > NoisyGL is indeed one of its first benchmark and very valuable one, but it is not applicable to our setting because it focuses exclusively on **node classification**, while all datasets, protocols, and GLN methods it standardizes operate at the node level. Our work, by contrast, studies **graph level classification** and relies on graph level Dirichlet energies, which cannot be computed or meaningfully evaluated on NoisyGL datasets. As the NoisyGL authors themselves say, **“ However, there is limited work on graph classification [25] and graph transfer learning [27] in the presence of label noise. Overall, research in other areas of graph learning, beyond node classification, is still in its early stages, and warrants further attention and exploration"**. This is exactly the gap our paper addresses.
> > >
> > > We appreciate your suggestion regarding reference formatting and will ensure all citations will be corrected in the final version.

---

### Official Review · Reviewer_xeHC · 2025-11-02

**Soundness:** 2
**Presentation:** 2
**Contribution:** 1
**Rating:** 2
**Confidence:** 4

**Summary:**

This paper studies the robustness of GNNs to noisy labels in graph classification tasks. Specifically, the authors investigate how GNNs leverage smooth representations as an inductive bias for generalization and noise robustness—an effect that can be disrupted by noisy labels. And then, three ways of avoiding GNN fitting on the noise are discussed and evaluated.

**Strengths:**

1. This work is the first to study the link between label noise robustness and the spectral dynamics of Dirichlet energy in graph classification.
2. It shows that three different approaches to improving robustness can be understood from an energy-based perspective that constrains harmful high-frequency energy.
3. Experimental results verify the effectiveness of the proposed GCOD loss.

**Weaknesses:**

1. Based on the definition of Dirichlet energy in Equation (2), the discussion in the paper is restricted to homophilic graphs, which makes the proposed improvement less broadly applicable.
2. A wider range of real-world datasets and a more detailed analysis of the results are required to validate the generality of the smoothness assumption and to clarify what the proposed loss function contributes to specific graph structures.

**Questions:**

See the weaknesses above.

---

> ### Author Response · Authors · 2025-11-20
>
> ### Weaknessess:
>
> 1.  Thank you for your comment we want to clarify that our setting is **graph classification**, not **node classification**. Therefore,    homophily/heterophily does not apply in the usual sense. Specifically, our work focuses on **graph classification**, where labels are assigned to **entire graphs**, not individual nodes.
>
>     Our method **does not assume homophily** of the graph structure, our **Dirichlet energy** ($Eq. 2$) measures **representation smoothness**, not label smoothness (not connected at all with the homophily/heterophily graphs definition).
>
>
>      An additional remark is that GCOD does not explicitly enforce smoothing, but encourages **representation consistency**; and in this case of graph classification this naturally leads to stable and low Dirichlet energy during training.
>
>
>      We agree that extending the method to **node classification**, where homophily/heterophily matter, would be valuable. Because GCOD does not rely on any homophily assumption, we expect it to generalize, and we consider this an exciting future direction.
>
>
> 2. We agree that wide coverage is important but our experiments already include datasets from **diverse domains and structural regimes**, covering:
>
>    * **Bio/chemical networks**: OGBG-PPA, PROTEINS, ENZYMES, MUTAG.
>    * **Social graphs**: IMDB-B, REDDIT-MULTI-12K.
>    * **Vision graphs**: MNIST-Graph.
>
>    These datasets vary in average graph size  (17 to 391 nodes) ,structural regularity (molecules, social networks, grid-like pixels), node-feature availability (present/absent) and graph density and topology
>
>     Across these heterogeneous settings, GCOD **consistently improves robustness** over CE and performs competitively with recent robust baselines (OMG, SOP (image baseline)), suggesting that the smoothness based perspective indeed captures a general phenomenon. We would gladly include more datasets. If the reviewer recommends specific ones, we are happy to add them in the revision.

---

### Author Response · Authors · 2025-12-02
**Response to the Area Chair**

We thank all reviewers for their feedback. We would like to clarify that the primary criticism underlying the lowest score stems from a misunderstanding of the task setting. Reviewer **xeHC** interpreted our method through the lens of node classification and concluding that the approach requires homophilic graphs. This is factually incorrect for our problem as our paper studies graph level classification, where homophily/heterophily are not defined, and our Dirichlet energy measures representation smoothness, not label smoothness. As a result, the central concern motivating this reviewer’s low score does not apply to our work.

Regarding Reviewer **WGPK** assessment, the observation that gains over **OMG** appear small overlooks the critical difference in methodological complexity. OMG requires dual graph augmentations, consistency regularization, and a mixing network, incurring roughly **200%** runtime overhead. In contrast, GCOD adds a single per-sample scalar correction, requires no augmentations or architectural changes, and introduces only **0.029%** overhead even on large datasets. Achieving comparable or superior robustness at a fraction of the cost is a meaningful and practically significant contribution for graph classification, where computational efficiency is essential.

The positive evaluations reflect the actual technical merit of the work. Reviewer **TcSw** highlights the novelty of connecting noisy label overfitting to Dirichlet energy dynamics and emphasises the strong empirical evidence and relevance of studying robustness in graph level tasks. Reviewer **5D5b** finds the paper clear, well motivated, and confirms that concerns about synthetic noise are fully resolved; as we clarified, real noisy graph-classification datasets do not exist, NoisyGL is node-level only, and our asymmetric noise protocol is structured and geometry-aware rather than random. Furthermore, GCOD remains inactive on clean labels, preserving clean-data performance and preventing over-regularization, consistent with our theoretical interpretation.

In summary, two reviewers provide strong, technically grounded support for acceptance, while the critical reviews are primarily based on misunderstandings rather than substantive flaws in the theory, methodology, or experiments. The paper offers a novel spectral perspective on robustness, introduces three principled and independent methods, and demonstrates practical value through highly efficient and stable performance across diverse graph datasets. We respectfully submit that, in view of the corrected interpretations and the aligned assessments of the positive reviewers, the evidence strongly favour acceptance.

---

### Meta-Review · Area_Chair_VJHp · 2026-01-07

**Summary:**

This paper explores the robustness of GNNs to noisy labels via an analysis of Dirichlet energy. While the empirical observation linking overfitting to increased Dirichlet energy is interesting, there are several issues.  Reviewers argue that the analysis is largely restricted to homophilic graphs and relies heavily on synthetic noise, limiting generality. The proposed methods yield only marginal gains over existing baselines, and key components such as the GCOD loss lack sufficient clarity and ablation.

**Reviewer Concerns:**

Reviewers have concerns about limitations on homophilic graphs and synthetic noise, limiting generality. In addition, the proposed methods yield only marginal gains.

**Reviewer Scores:**

Reviewers generally keep their scores after rebuttal.

---

### Decision · Program_Chairs · 2026-01-26

Reject